https://doi.org/10.1038/s41467-020-20084-5　**OPEN**

# Direct observation of the formation and stabilization of metallic nanoparticles on carbon supports

Zhennan Huang [1,9], Yonggang Yao [2,9], Zhenqian Pang[3,9], Yifei Yuan[1,4], Tangyuan Li[2], Kun He[5], Xiaobing Hu [5], Jian Cheng [3], Wentao Yao [6], Yuzi Liu [7], Anmin Nie[8], Soroosh Sharifi-Asl [1], Meng Cheng[1], Boao Song [1], Khalil Amine [4], Jun Lu [4], Teng Li [3✉], Liangbing Hu [2✉] & Reza Shahbazian-Yassar [1✉]

Direct formation of ultra-small nanoparticles on carbon supports by rapid high temperature synthesis method offers new opportunities for scalable nanomanufacturing and the synthesis of stable multi-elemental nanoparticles. However, the underlying mechanisms affecting the dispersion and stability of nanoparticles on the supports during high temperature processing remain enigmatic. In this work, we report the observation of metallic nanoparticles formation and stabilization on carbon supports through in situ Joule heating method. We find that the formation of metallic nanoparticles is associated with the simultaneous phase transition of amorphous carbon to a highly defective turbostratic graphite (T-graphite). Molecular dynamic (MD) simulations suggest that the defective T-graphite provide numerous nucleation sites for the nanoparticles to form. Furthermore, the nanoparticles partially intercalate and take root on edge planes, leading to high binding energy on support. This interaction between nanoparticles and T-graphite substrate strengthens the anchoring and provides excellent thermal stability to the nanoparticles. These findings provide mechanistic understanding of rapid high temperature synthesis of metal nanoparticles on carbon supports and the origin of their stability.

[1] Department of Mechanical and Industrial Engineering, University of Illinois at Chicago, Chicago, IL 60607, USA. [2] Department of Materials Science and Engineering, University of Maryland, College Park, MD 20742, USA. [3] Department of Mechanical Engineering, University of Maryland, College Park, MD 20742, USA. [4] Chemical Sciences and Engineering Division, Argonne National Laboratory, Lemont, IL 60439, USA. [5] Northwestern University Atomic and Nanoscale Characterization Experimental (NUANCE) Center, Northwestern University, Evanston, IL 60208, USA. [6] Department of Mechanical Engineering-Engineering Mechanics, Michigan Technological University, Houghton, MI 49931, USA. [7] Center for Nanoscale Materials, Argonne National Laboratory, Lemont, IL 60439, USA. [8] Center for High Pressure Science, State Key Lab of Metastable Materials Science and Technology, Yanshan University, Qinhuangdao 066004, China. [9] These authors contributed equally: Zhennan Huang, Yonggang Yao, Zhenqian Pang. ✉email: lit@umd.edu; binghu@umd.edu; rsyassar@uic.edu

Substrate-supported nanoparticles are of great interest to the industry due to their broad applications in biology-medicine, energy storage and catalysts[1–4]. Carbon is the most commonly used conductive substrate and is naturally abundant as well. This material has a variety of unique morphologies ranging from zero to three dimensional[5], which could be used to anchor nanoparticles and scale up to form desired structures. To date, various methods have been developed to synthesize carbon-supported nanoparticles[6–9], although, it remains challenging to achieve nanoparticles with uniform size and dispersion. One can realize this goal by introducing dispersion agents or surfactants[10–12]. However, the side effects of the residues from solution-based synthesis can be problematic[13–16]. Recently, high temperature dry synthesis methods[17–20] have been successfully introduced to produce nanoparticles, including pure metals, multicomponent alloys and even single atoms. In this method, no additive is required during the synthesis process, which not only reduces the complexity of the synthesis but also enables "clean" synthesis strategies.

Very recently, we reported a fast, high temperature, and dry synthesis method based on rapid Joule heating of a carbon matrix loaded with metal salt precursors (carbothermal shock)[20]. This process generates extremely high temperature (>1500 K) to thermally decompose the loaded salt precursors to metallic nanoparticles[20–22]. The synthesized nanoparticles are well dispersed on carbon substrate with a high loading rate and good size controllability. In addition, by mixing various metal salt precursors, we could obtain multicomponent nanoalloys which is extremely promising for the next generation of catalyst materials.

In spite of versatility and simplicity of high temperature Joule heating method, the underlying mechanisms behind the formation of small and well-dispersed nanoparticles without aggregation at high temperature processing are not well understood. A few atomistic simulations have attempted to study the interaction between the metallic nanoparticles and graphitic basal planes[23,24] or defective basal planes[25–27]. However, the effect of high temperatures is not considered in those simulations. This is even more important for this case, since at such high temperatures in Joule heating process, one would expect that the nanoparticles become agglomerated to reduce their surface energy[17]. Nevertheless, there is a lack of understanding about the stability of nanoparticles on carbon substrates at elevated temperatures.

Nanoscale in situ characterization techniques, especially in situ transmission electron microscopy (TEM) has shown the ability to monitor the kinetic process on various nanoscale materials at unprecedentedly high spatial resolutions[28–31]. In this work, we utilize electrical-biasing in situ TEM setup to mimic the high temperature shock approach and investigate the formation and stabilization of nanoparticles on carbon supports during this process. The carbon nanofibers (CNFs) are first immersed in metal precursor solutions and then subjected to Joule heating inside TEM. We observed that the formation of metallic nanoparticles was associated with the simultaneous crystallization of CNFs. Our results suggest that the phase transition of CNF results in the formation of a disorder turbostratic graphite (T-graphite) structure. In addition, the fast gas evolution during phase transition generates T-graphite structure with highly defective edge planes. Through high resolution TEM (HRTEM) and atomistic simulations, we find that the defective edge planes are the preferred sites for the formation and stabilization of metal nanoparticles. The carbon defects provide the ideal nucleation sites and the formed nanoparticles further take root on edge planes through the intercalation of metal atoms between graphite planes, resulting in even stronger binding between nanoparticles and carbon support. The high binding energy helps to stabilize the nanoparticles on the carbon substrate at extremely high temperatures. Further in situ annealing studies up to 1173 K also

confirms the excellent thermal stability of nanoparticles. We believe the mechanistic details reported in this work would not only guide the nanoparticles synthesis through Joule heating-based methods, but also pave the way on understanding other rapid high temperature synthesis approaches.

## Results

**In situ TEM visualization of Joule heating on metal salt loaded CNF.** In situ Joule heating experiment is proceeded by utilizing an in situ TEM electrical-biasing holder. As schematically shown in Fig. 1a, gold (Au) and tungsten (W) rods are used as two electrodes. To induce the Pt nanoparticles formation, the $H_2PtCl_6$ salt precursor loaded CNF (S-CNF) is connected across the two electrodes and subjected to Joule heating. The morphological changes in the S-CNF due to Joule heating is visualized in the form of a series of bright field TEM images and Movie S1. Interestingly, we noticed that upon Joule heating, the S-CNF suddenly expanded, and many dark-contrast nanoparticles formed on the CNF. This is demonstrated in Fig. 1b, where a S-CNF with an average diameter of ~250 nm is shown to be placed between two electrodes. After applying electrical current, the diameter of the CNF rapidly increases to ~330 nm and nanoparticles form on the CNF substrate (Fig. 1c, d). The schematic in Fig. 1e represents a summary of the observation where the diameter of S-CNF increases simultaneous to the formation of nanoparticles on CNF substrate.

Figure 2 depicts the details of structural transition of S-CNF due to Joule heating. As shown in Fig. 2a, the surface of a pristine S-CNF is covered with a layer (marked with dashed yellow line) representing the coated $H_2PtCl_6$ salt layer. After applying current, the CNF swells and turns into wrinkled structure along with the formation of nanoparticles (Fig. 2b, $t = 0.03$ s). This Joule heating process is shown in Movie S2. In order to capture this process at higher image recording speeds, the in situ Joule heating experiments were repeated on a TEM equipped with ultrafast CMOS camera at an image acquisition speed of ≥200 fps achieving less than 5 ms time resolution (Movie S3 and S4). The Joule heating-induced structural evolution showed similar behavior, as shown in Figs. S1 and S2, which means that the phase transition and the formation of nanoparticles occur at temporal rates faster than 5 ms. The phase evolution of S-CNF is analyzed by selected area electron diffraction (SAED). The pristine S-CNF is amorphous, as shown by the broadly diffusive rings in the SAED pattern (Fig. 2c). After Joule heating, the amorphous carbon turns to a polycrystalline structure, as indicated by the sharp rings in Fig. 2d. The outmost two rings (yellow dashed curves) are measured to be 2.12 Å and 1.23 Å, close to (100) and (110) planes of graphite ($P6_3/mmc$; $a = b = 2.470$ Å, $c = 6.724$ Å; JCPDS 41-1487), respectively. Interestingly, the inner yellow dashed ring is identified to be 3.46 Å, which is much larger than the interplanar distance of a typical graphite basal plane (3.35 Å). However, it matches well with the disordered crystalline (turbostratic) carbon structure (T-graphite)[32]. The nanoparticles are confirmed to be metallic Pt, since the $d$ spacing of three dashed red rings are measured to be 2.26 Å, 1.96 Å and 1.38 Å, respectively. This matches well with the interplane distance of (111), (200) and (220) planes of FCC Pt crystal structure ($Fm-3m$, $a = b = c = 3.923$ Å, JCPDS 04-0802), and indicates the decomposition of precursor salt to metallic nanoparticles during this high temperature process. Thus, we can conclude that the Joule heating method lead to the simultaneous crystallization of amorphous CNF support and Pt nanoparticles formation.

To compare with other heating methods, we performed in situ heating experiments on S-CNFs utilizing microfabricated-heating SiN membrane systems. No graphitization or expansion of

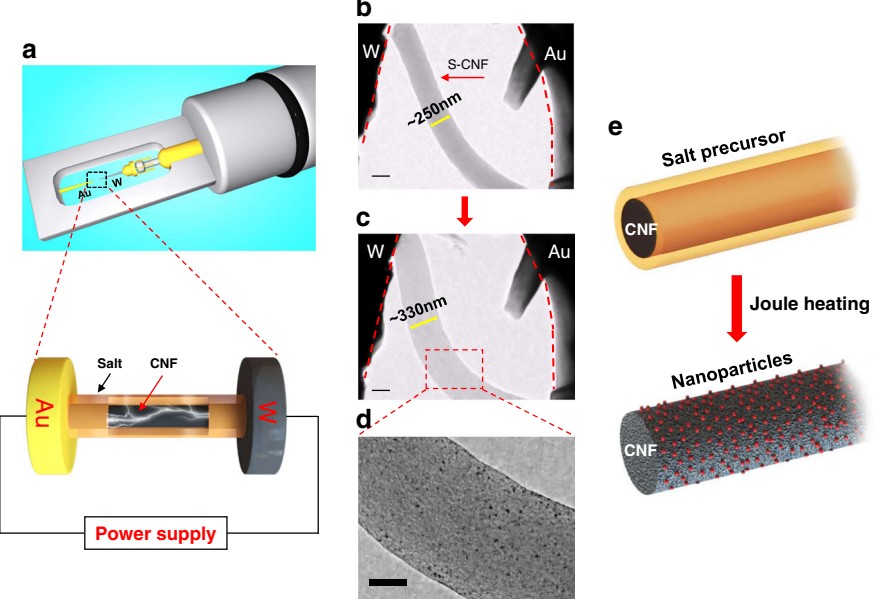

**Fig. 1 In situ TEM observation of Pt nanoparticles formation on H₂PtCl₆-loaded CNF through electrical Joule heating. a** Schematic images of electrical biasing TEM holder and the zoomed-in view of the salt-loaded nanofibers between Au and W electrodes. **b–d** TEM images depict the CNF expansion and the formation of Pt nanoparticles on CNF. Scale bars are 200 nm in (**b**, **c**), 100 nm in (**d**). **e** The schematic of the S-CNF before and after the Joule heating.

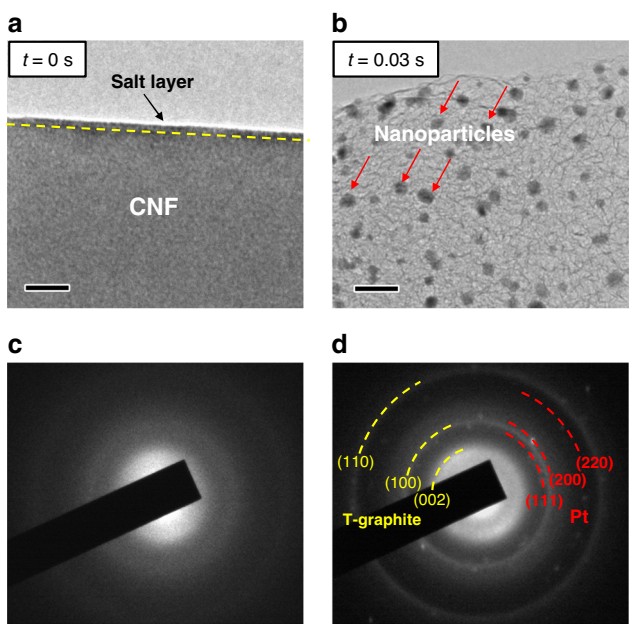

**Fig. 2 Structural evaluation of salt-loaded amorphous CNF during Joule heating process. a**, **b** Bright field TEM images of the S-CNF at $t = 0$ s and $t = 0.03$ s, respectively, and **c**, **d** the SAED patterns of pristine S-CNF and Joule heated S-CNF, showing the crystallization of CNF and the formation of metallic Pt nanoparticles, respectively. Scale bars are 20 nm.

amorphous carbon substrate was observed. The Pt nanoparticles appear at temperatures between 473 to 673 K (Fig. S4b) but they continually grow to larger sizes (Fig. S4c–S4f) over time. This probably could be related to the slower heating rate in the membrane devices versus the Joule heating. Such slow heating rate can induce metal salt dehydration/decomposition at lower temperatures leading to metal nanoparticles formation without graphitization (Movie S5). Due to slower rate of heating rate, it is

challenging to obtain multielement alloyed nanoparticles without chemical segregation (Fig. S4g).

The crystallization of amorphous carbon to T-graphite may be induced either by noble metals, because of their high catalytic activity[33,34], or the extremely high temperatures[35,36] generated by Joule heating. To differentiate the above two scenarios, pristine amorphous CNFs without salt precursor coating were used to run the Joule heating process. Figure 3a shows a pristine CNF where the surface is relatively smooth. The SAED pattern in the inset of Fig. 3a indicates that the pristine CNF has an amorphous structure. Upon Joule heating, this CNF shows similar expansion and wrinkled structures as observed previously in Fig. 1 and Fig. 2. The TEM image (Fig. 3b) and the corresponding SAED image (inset in Fig. 3b) also depict a polycrystalline structure and the diffraction rings matches well with T-graphite planes. The Joule-heated CNF is further analyzed by HRTEM, where the wrinkled structure is clearly shown (Fig. 3c). The wrinkled structures are identified as the edge planes of T-graphite, matching the *d* spacing of 3.46 ± 0.01 Å. Thus, one can conclude that the main factor leading to the CNF graphitization is the high temperature effect rather than the catalytic effect of Pt nanoparticles.

Quantitively analysis of the CNF composition before and after Joule heating is performed by energy-dispersive X-ray spectroscopy (EDS). As shown in Fig. 3d, significant amount of noncarbon elements, 13.0 wt% nitrogen (N) and 4.3 wt% oxygen (O) are detected in pristine CNF (corresponding EDS spectrum Fig. S3a), which is caused by the relatively low carbonization temperature during preparation[37]. However, after Joule heating, the amount of N and O elements drop significantly to 2.0 wt% and 1.3 wt%, respectively (corresponding EDS spectrum Fig. S3b, the conversion of at% to wt% was applied). Since noncarbon elements in the CNF are volatilized in the form of various gases at high temperatures[35,36], the significant reduction of N and O elements indicate the release of $N_2$, CO and other gases[36,38,39]. As depicted in Fig. 3e, the removal of O as CO or $CO_2$ creates defects on the benzene ring-like chains[40]. While the removal of N as $N_2$ would crosslink the adjacent chains to enlarge the carbon basal

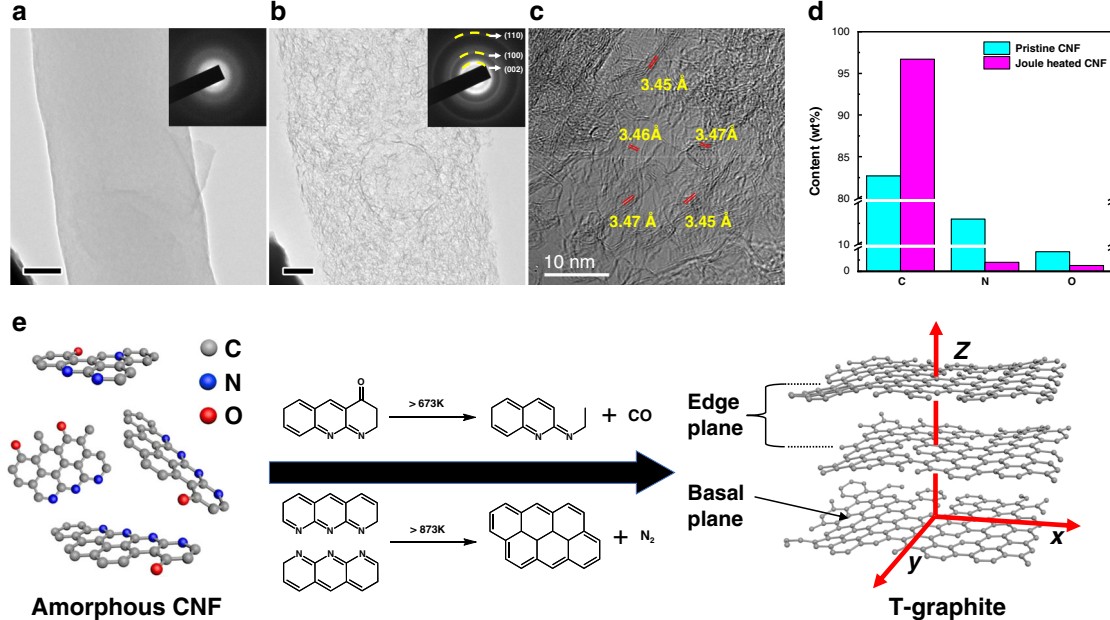

**Fig. 3 TEM and EDS analyses of pristine amorphous CNF evolution during Joule heating. a** TEM image of original CNF without salt loading is shown. The corresponding SAED pattern displays diffusive rings, which is a characteristic of an amorphous structure. Scale bar is 50 nm. **b** TEM image of wrinkled CNF achieved after Joule heating, where the SAED pattern shows sharp rings, corresponding to T-graphite. Scale bar is 50 nm. **c** HRTEM image of the wrinkled structure and **d** quantitative EDS analysis of carbon, nitrogen and oxygen content in a CNF before and after Joule heating. **e** Amorphous CNF evolution during Joule heating and typical routes of N and O removal at high temperature.

plane[41]. The significant drop of N and O elements indicates a large amount of gas release, which explains the expansion of CNF (Movie S1 and S2). After the removal of the majority of N and O elements, the carbon (C) content increases from 82.7 wt% to 96.7 wt%. It has been reported that CNF could achieve 96.0 wt% carbon content at 1573 K by slow heating in conventional furnaces (ramping rate 513 K/h)[35,40]. The Joule-heated CNF achieves higher carbon content in milliseconds level, which means the temperature of Joule heating process is much higher than 1573 K. As a conclusion, the CNF evolution mechanism at high temperatures is schematically shown in Fig. 3e. The pristine CNF consists of nonordered carbon nanostructures with hetero N and O atoms. By expelling these hetero atoms in the form of gases at high temperatures, the carbon nanostructures crosslink together to enlarge the carbon basal plane and form defects on the edges. However, since a small amount of noncarbon atoms still remains and the period of high temperature is relatively short, the amorphous CNFs are not transformed to fully crystalline graphite but rather an intermediate T-graphite structure form.

**Finite element analysis of temperature evolution on CNF during Joule heating**. The temperature evolution on the CNF during Joule heating was mapped through finite element analysis (FEA). As demonstrated in Fig. 4a, a CNF is bridged between the Au and W electrodes. After applying the input power ~40 μW (same as in situ experiments), the center areas of the CNF reach to a thermal equilibrium at ~2000 K. More precisely, the temporal evolution of temperature at the center point of CNF is shown in Fig. 4b. The temperature rapidly increases to ~1600 K within 4 μs, and eventually reaches to an equilibrium temperature of ~2000 K at 10 μs. Based on the Joule's first law, heat conduction always exists. Thus, the areas close to the electrodes show a temperature gradient towards lower temperature. The temperature evolution of the entire CNF is visualized in Movie S6.

**Structural evolution of CNF and nanoparticles formation during Joule heating**. With the mechanistic clarification of the CNF evolution, the nanoparticles formation during Joule heating now can be explained. Normally at elevated temperatures, metal nanoparticles are expected to agglomerate into large particles to reduce their surface energy[6,42,43]. However, in our case, after exposure to extremely high temperatures during Joule heating, the formed nanoparticles are very small in size and well separated on the carbon support. As seen in in Fig. 5a, the Joule heated carbon substrate is composed of randomly oriented edge planes. Interestingly, it is found that the nanoparticles are mostly located at the wrinkle-like edge planes as marked with red arrows, while fewer nanoparticles exist at the smooth carbon basal planes (dashed yellow areas). This is schematically showing in Fig. 5b as well. In addition, the edge layers underneath the nanoparticles are largely expanded and distorted as shown by the inset image of Fig. 5a. Overall, it appears that the edge planes of T-graphite are the preferable sites for the nanoparticles to nucleate and become stabilized.

To determine if our observation is limited to Pt nanoparticles chemistry, we performed Joule heating on CNFs coated with $RuCl_3$ and the mixture of $H_2PtCl_6$, $PdCl_2$, and $NiCl_2$ precursors as well. Interestingly, similar morphologies of nanoparticles and carbon support was observed in this case. As shown in Fig. 5c, d, single component ruthenium (Ru) as well as ternary PdPtNi nanoparticles are synthesized and resolved by annular bright field (ABF)-scanning TEM (STEM) imaging. The Ru and PdPtNi nanoparticles are formed on edge planes as indicated by the ABF-STEM images (Fig. 5c, d) and the composition is confirmed by the EDS mapping in Fig. 5d and f, respectively.

In order to rule out the effect of electromigration on the formation of nanoparticles, rapid radiative heating process was used to synthesize nanoparticles on carbon support. The rapid radiative heating setup is schematically shown in Fig. S5a. In this configuration, the outside carbon film acts as a source for flash high temperature radiation to heat up the S-CNF. At this

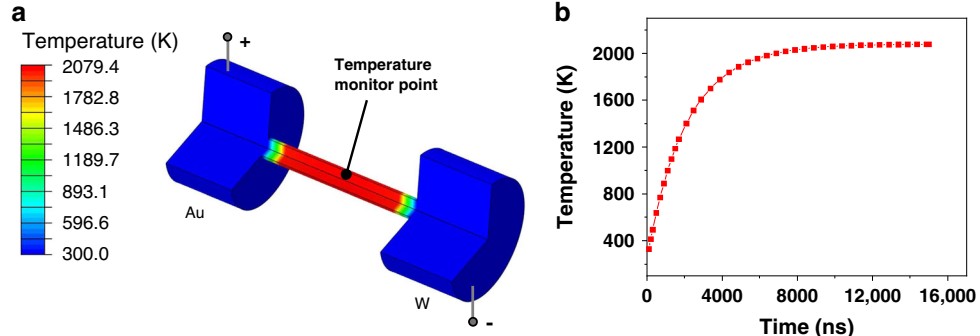

**Fig. 4 Finite element analysis of the Joule heating process of a carbon fiber upon applying the input power of ~40 μW. a** Temperature profile at 10 μs when the system has reached a thermal equilibrium of ~2000 K. For clarity, a quarter of the model is removed from the result and the electrodes are partially shown. **b** Temporal evolution of the temperature at the center monitor point (shown in **a**) inside the CNF.

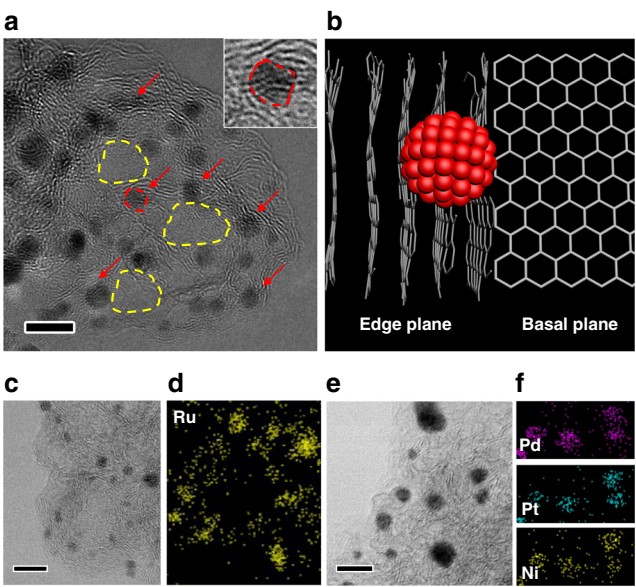

**Fig. 5 The HRTEM and STEM characterization of single element and multielement nanoparticles on the Joule heated S-CNFs. a** The HRTEM image of Pt nanoparticles located on edge planes (marked by the red arrows). The carbon basal planes are marked as the yellow dashed circles. The inset shows an enlarged view of a nanoparticle (marked by the red dashed area). **b** Schematic represents a nanoparticle that is reside on edge plane instead of basal plane in T-graphite. **c** The ABF-STEM image shows Ru nanoparticles on edge planes, and **d** the corresponding EDS mapping. **e** The ABF-STEM image of ternary PdPtNi nanoparticles on edge planes and **f** the corresponding EDS mapping. All scale bars are 5 nm.

condition, the salt-loaded CNFs were only exposed to high temperatures under ultrafast heating rate ($>10^5$ K/s), and no direct electrical current was applied to the S-CNFs. After rapid radiative heating treatment, the prepared Pt NPs@CNF specimens were analyzed via HRSTEM. As we can see from Fig. S5b, a uniform distribution of small nanoparticles was observed on the CNF support. Similar to the Joule heating case, the amorphous carbon film showed evidence of crystallization and the NPs were associated with the T-graphite edge planes. To summarize, we included the above experiments in Table S1. Overall, it is evident that the formation and stabilization of nanoparticles are governed by the rapid heating process, and not necessarily electromigration.

To investigate the underlying mechanism of edge planes affecting the formation and stabilization of nanoparticles during ultra-high temperature Joule heating, density of functional theory (DFT) and large-scale molecular dynamics (MD) simulations were carried out. The edge planes form with a highly defective nature (full of carbon dangling bonds) as discussed previously and reported in the literature[11,44]. Thus, we calculate the binding energy of Pt and carbon with dangling bond (details are in the "Methods" part) and the cohesive energy of Pt cluster through DFT method. The binding energy of Pt and carbon ($-8.51$ eV/atom) is significantly higher than the cohesive energy of Pt cluster ($-5.37$ eV/atom). In other words, at high temperatures, instead of joining the adjacent clusters to form larger nanoparticles, Pt prefers to distribute in the areas containing carbon dangling bonds, especially the edge planes of T-graphite. To further analyze the thermal stability of the Pt nanoparticles on edge planes, we model a Pt cluster with 100 Pt atoms on the edge of T-graphite (Fig. 6a) and heat the entire system up to 1800 K. Interestingly, the Pt cluster at such high temperature well maintains its location and a few atoms of the cluster tend to intercalate into the layers (Fig. 6b). The intercalated atom is relatively stable on the edge plane by our MD calculations (Fig. S6). As the relaxation continues, the cluster adhered on the edge plane tightly as shown in Fig. 6c. Meanwhile, in agreement with the observed interlayer expansion shown in the inset of Fig. 5a, the intercalated Pt atoms increase the interlayer distance of graphite sheets, as shown by the change of spacing between white line segments from nearly parallel ($t = 0$) to largely expanded ($t = 60$ ps) layers (Movie S7). When the Pt cluster adheres to the basal plane of T-graphite (Fig. S7), it can be found that the cluster moves around and cannot pin on the plane stably. Figure S8 shows the potential energy of the Pt particle as the function of relaxation time in the presence of T-graphene with basal plane and edge plane. The potential energy of the Pt particle on the basal plane keeps decreasing gradually, which means the particle is not stable. To further demonstrate the stability on the edge plane of graphene, we also calculated the charge density by DFT method (Fig. S9). Figure S9b shows the charge density difference of Pt on the basal plane. There exists few charge transfer between the Pt particle and the graphene plane, which suggests that the main interaction between the Pt particle and the graphene basal plane is van der Waals force and it is hard to form the covalent bond between these two atoms. On the contrary, large amount of charge transfer occurs between the Pt particle and the graphene edge plane (Fig. S9d), which suggests strong binding energy between the Pt particle and the edge plane of graphene (in agreement with MD simulation results in Fig. S8). The above two types of simulations clearly show that the diffusion of the Pt particle on the edge plane is much harder than that on the basal plane. Considering the different orientation of graphene

edges, we compared the charge densities of Pt particles on both armchair and zigzag edges (Fig. S10). There exists large amount of electron transfer in both cases, which means the main interaction is strong covalent bonding rather than the van der Waals force. Furthermore, three distinct regions are considered to analyze the stability of Pt. Figure S11a shows the isolated Pt atoms intercalated far from the edge plane. The Pt atoms prefer to locate on the bridge site of graphene. Similar to the Pt particle on the basal plane, there exists few charge transfer between Pt and C atoms (Fig. S11d), while the Pt atoms at the edge plane have the regular charge transfer and are bonded with graphene through covalent bonds. Thus, it forms the graphene-Pt alloy (Fig. S11b

and e). When the Pt particle is attached to the Pt-rich edges (Fig. S11c, f), the charge transfer becomes irregular and the change of charge density leads to the stability of particle at the edge plane.

**Computational studies on the behavior of Pt nanoparticles on CNFs.** In order to illustrate the temperature effects on the stability of Pt cluster, the binding energy in cluster/T-graphite is calculated at 300 K and 1800 K. Figure 6d shows that the cluster has three to five times higher binding energy to the edge plane than on basal plane. The higher temperatures facilitate the incorporation of Pt atoms into the graphite interlayers resulting in higher binding energy and excellent stability on edge planes at high temperatures. Lower temperature (1000 K) condition is also simulated (Fig. S12), which shows similar phenomenon. In order to understand the effect of edge planes thickness on the mobility of Pt cluster, we evaluated the T-graphite with more layers (fifteen layers). Similar to the previous case, the Pt cluster is tightly bonded with the T-graphite edges (Fig. S13). Compared with the case of 300 K, more Pt atoms are inserted into the edge interlayers at 1800 K temperature (Movie S8 and S9).

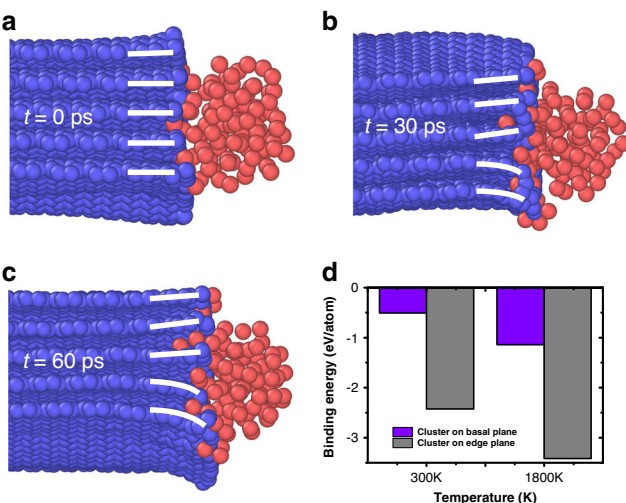

**Fig. 6 The morphological derivation of Pt cluster in the presence of T-graphite with edge planes. a–c** The morphology of Pt cluster residing in edge planes with the temperature 1800 K, after different time relaxations. **d** The dependence of binding energy of cluster with different locations and various temperatures.

**In situ thermal stability analysis on Joule heated Pt-CNF.** The MD simulations indicates that the nanoparticles are quite stable on the edge planes of T-graphite even at extremely high temperatures. To further verify the thermal stability of such nanoparticles, we performed in situ TEM heating on the CNFs decorated with Pt nanoparticles that were synthesized through ex situ Joule heating method. The ex situ formed nanoparticles appear to have very similar morphology to the in situ observations reported in this work. As it can be seen in Fig. 7a, the ultrafine nanoparticles are located at wrinkled edge planes showing FCC structure (inset of Fig. 7a). The Pt-decorated CNFs are evaluated at various temperatures (573 K, 773 K, 973 K, and 1173 K). As shown in the high angle annular dark field (HAADF)-STEM image in Fig. 7b, the ultra-small Pt nanoparticles are well

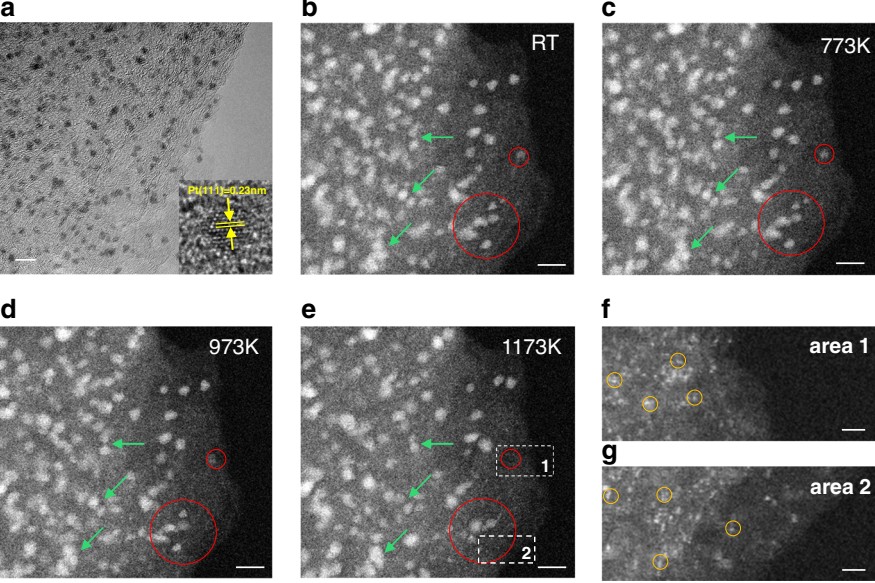

**Fig. 7 Thermal stability studies of Pt nanoparticles on CNF supports through in situ annealing. a** Large scale TEM image of nanoparticles sitting on wrinkled edges (inset shows HRTEM image of a Pt cluster). HAADF-STEM images of Pt nanoparticles on CNF substrate captured after annealing at different temperatures: **b** RT, **c** 773 K, **d** 973 K, and **e** 1173 K. The particles are highly stable with no obvious aggregation. Red circles in each image depict nanoparticles that mostly maintain their stability during heating from RT to 973 K, while some of the nanoparticles vanish at 1173 K. Scale bars are 5 nm. The images in (**f**, **g**) show the high magnification view of areas 1 and 2 (corresponding to the areas marked in **e**), where nanoparticles disappear. Interestingly, high population of single atoms (some of them are indicated by the yellow circles) are observed. Scale bars in f and g are 2 nm.

dispersed on CNF at room temperature (RT). High temperatures are expected to be detrimental to the nanoclusters' stability, since the nanoparticles would fuse into larger particles to reduce the surface energy at elevated temperatures[17]. However, by increasing temperature, the nanoparticles, even those at very close distances to each other (shown by green arrows), do not show obvious aggregations up to 1173 K (Fig. 7b–d). Thus, the excellent thermal stability of the nanoparticles is confirmed by the in situ heating studies. Unexpectedly, some of the nanoparticles disappear at 1173 K (two of those areas are marked with red circles). To clearly this, a close examination of all the nanoparticles is performed by comparing the TEM images taken at RT and at 1173 K. As shown in Fig. S14, all the outlines of the nanoparticles were marked (Fig. S14b, d) and overlapped (Fig. S14e). The detailed statistical analysis is shown in Tables S2 and S3 and summarized here: (1) about 78% of the nanoparticles barely moved, and still largely overlapped with the RT locations after 1173 K treatment; (2) about 16% of the nanoparticles were disappeared; (3) only about 4% of new nanoparticles appeared (possibly due to Ostwald ripening) and 3% of particles moved (possibly due to lose anchoring to the edge planes); and (4) the average size does not change (3.2 nm vs 3.2 nm), while the total area of the nanoparticles reduced by 12.4% (from 280.5 to 245.6 nm$^2$). Since the size of the nanoparticles at 1173 K does not increase, the agglomeration of nanoparticles can be ruled out. To better understand this, two locations (marked by the white dashed frames and labeled as 1 and 2 in Fig. 7e) are chosen for further analysis. Interestingly, in these two locations we notice the presence of many single or few atoms as shown in Figs. 7f and g (some are marked with yellow circles). Considering that in our case, strong bonding exists between the T-graphite edge planes and nanocluster, the mobility of nanoparticles should be highly restricted. However, individual atoms should gain sufficient momentum to break the Pt–Pt metallic bonds, resulting in the detachment from the cluster. Since the substrate is highly defective, the detached atoms can re-bond with carbon defects. As a consequence, the atomically distribution of Pt is seen in areas 1 and 2.

Based on our MD simulation, some of the atoms intercalate into graphene layers and bond on the carbon substrate, which may result in the disappearance of nanoparticles. It is more likely that the nanoparticles convert to single atoms and disperse on the substrate as discussed here. To verify this scenario, we also calculated the Pt particle with smaller sizes, which is constructed with 12, 25, 50, and 75 atoms, adsorbed on the edge plane of graphene (Fig. S15). It can be found that the $Pt_{12}$, $Pt_{25}$, and $Pt_{50}$ disappeared and the Pt atoms intercalated into the graphene layers, while $Pt_{75}$ is stable after parts of atoms intercalated into graphene layers. It can be concluded that there are not enough Pt–Pt bonds in the small Pt particle to resist the Pt intercalation, leading to the disappearance of Pt particle. When the particle is large enough (number of atoms ($n$) in our simulation is larger than 75), remaining Pt–Pt bonds are strong enough to resist further intercalation. Fig. S15e shows the potential energy of Pt particles as the function of relaxation time. All these curves converge to a constant. The smaller the particle is, the more quickly the curve converges. It also means that the intercalated Pt atoms are stable in the graphite layers, even the particles that disappeared ($n < 75$). On the other hand, it confirmed that some small particles disappeared, and only large particles can be maintained in our experiments. This is in agreement with a recent report[17] showing the conversion of nobel metal nanoparticles to single atoms above 1173 K. Some other studies of noble metals on carbon-based substrates at high temperatures showed that the metal nanoparticles were not stable on substrates at less than 673 K[17,45,46]. Considering that the nanoparticles synthesized via Joule

heating showed improved stability by more than 300 K, we can safely conclude that the Joule heating synthesis of nanoparticles on amorphous carbon supports can yield ultra-stable nanoparticles.

## Discussion

Our work provides direct visualization of nanoparticles on amorphous carbon supports and elucidate the details of amorphous carbon graphitization during ultrafast exposure to extremely high temperatures under electrical Joule heating process. The in situ TEM studies of Joule heating on amorphous carbon fibers loaded with metal precursors show that during the Joule heating process, the CNFs undergo volumetric expansion due to amorphous to crystalline phase transition, which is also associated with formation and stabilization of metallic nanoparticles on the carbon supports. The crystallized CNFs are made of T-graphite with a high density of randomly dispersed edge planes. These edge planes are highly defective and act as preferred sites for the nucleation of metallic nanoparticles. In addition, metal atoms diffuse into layered structure strengthening the anchoring of the nanoparticles on the edge planes. This prevents the nanoparticles from further aggregation at extremely high temperatures as confirmed by MD simulation and in situ heating in TEM. In this work, while Pt nanoparticles are described as a model synthesis system, the same observation is made for Ru and multicomponent PdPtNi nanoparticles as a generalization. Overall, this work reveals new insights on mechanistic of Joule heating method and rapid high temperature synthesis of nanoparticles on carbon supports. In addition, the knowledge learned from this study can facilitates the synthesis of highly stable supported nanoparticles for applications geared toward environmental remediation (e.g., water or filtering) or energy storage and conversion (e.g., catalysis engineering).

One should note that the fabrication of nanoparticles by Joule-heating method is highly scalable, and the process is not limited only to single CNFs. In fact, large sheets of CNF films can be subjected to Joule heating (specimen dimensions from nm (single CNF) to mm level (CNF film) as shown in Fig. S17. This process can be further scaled up by connecting several CNF films in series where electrical current pass through the films[47]. In addition, by switching the carbon substrate to a bulk carbonized wood[20], the scalability increases to tens of centimeters. A recent study[48] also demonstrated that rapid Joule heating method is highly scalable, where gram level graphene/graphite material was synthesized through ultrahigh temperature Joule heating in milliseconds from amorphous carbon sources.

## Methods

**Sample preparation**. The pristine CNFs are prepared by electrospinning an 8 wt% polyacrylonitrile (PAN) in dimethylformamide (DMF) solution to receive nanofibers to receive polymer nanofibers. The nanofibers are then oxidized in air at 533 K for 5 h and carbonized at 1073 K for another 2 h to become CNFs. In order to synthesize metallic nanoparticles, a precursor solution is prepared by dissolving certain amount of salt precursor(s) ($H_2PtCl_6$, $RuCl_3$, $PdCl_2$, $NiCl_2$ (Sigma Aldrich)) in ethanol. The CNFs are thereafter soaked in the salt-ethanol solution for one hour and dried in air to achieve the salt-coated carbon nanofibers (S-CNFs). The ex situ sample was prepared through Joule heating of salt loaded CNF film (schematically shows in Fig. S16), where the CNF film image can be seen from Fig. S17 and the loading amount of Pt salt precursor was 0.25 µmol/cm$^2$.

**In situ TEM methods and characterizations**. TEM characterization is performed by JOEL 3010 and STEM imaging is carried out by JEOL ARM 200CF equipped with HAADF, ABF and EDS detectors. In situ TEM Joule heating setup is prepared by placing one terminal of S-CNF to an Au tip electrode in a Nanofactory electrical biasing TEM specimen holder. The other terminal of the S-CNF faces a W wire as the counter electrode. Piezo control of the specimen holder is used to move and connect the W electrode with the S-CNF to construct the whole circuit. In situ TEM thermal stability study is performed by using Protochips Aduro heating holder, with the sample loaded on the matched E-chip. The temperature is raised at

a heating rate of 50 K/s. Each temperature is kept for at least 30 min and then cool down to RT to take images.

**MD simulations.** Large-scale MD simulations using the massively parallel simulator (LAMMPS)[41] is performed to understand the interaction between Pt cluster and graphene films. The ReaxFF potential[49] is used to represent the interaction among carbon and Pt atoms in Pt/graphene system, which can accurately describe the structural properties of Platinum cluster and carbon planes and is broadly accepted in such similar calculation[49]. Time step is set as 0.25 fs and a Nose-Hoover thermostat is used to maintain the NVT ensemble for up to 120 ps (Fig. S18), which is long enough to analyze the mechanism[50,51]. Periodical boundary condition in three directions is employed to all these samples and 100 Pt atoms are assigned to construct the cluster. The energy of system is minimized until the total atomic forces is converged to less than $10^{-9}$ eV/Å. The content of other elements in our experiments is very low (Fig. 3d). Thus, the functional groups were not considered in our MD simulation. We set the initial temperature of the Pt/graphene system at 5 K to equilibrate the system and then heating it to room temperature (300 K) and 1800 K to analyze the thermal stability. After relaxing the Pt/graphene system sufficiently, we implement the static calculation to analyze the interaction between Pt cluster and graphene sheets. The binding energy of the Pt cluster to graphene is defined as:

$$E_{\text{binding}} = \frac{1}{n}\left(E_{\text{system}} - E_{\text{cluster}} - E_{\text{T-graphite}}\right)$$

where $E_{\text{system}}$, $E_{\text{cluster}}$ and $E_{\text{T-graphite}}$ represent the average potential energies of the system, isolated Pt cluster and T-graphene, respectively. $n$ is the number of Pt atoms in cluster.

**DFT simulations.** In order to obtain the cohesive energy of Pt cluster and the binding energy of Pt and carbon with dangling bond accurately, DFT implemented in the Vienna ab initio simulation package (VASP)[42,43] is employed. The generalized gradient approximation (GGA) of the Perdew-Burke-Ernzerhof (PBE) functional[44] are used for exchange and correlation interaction. For the cohesive energy calculation, we built the Pt cluster with 16 atoms and then delete one atom at the surface randomly. The box size is set as 20 Å × 20 Å × 20 Å to eliminate the interaction between periodic images of atoms. To the case of binding energy of Pt atom and carbon with dangling bond, we put one Pt atom on the top of defective graphene with monovacancy and set a vacuum space of 15 Å perpendicular to the graphene plane. The Brillouin zone in all these cases is sampled by a $1 \times 1 \times 1$ Monkhorst and Pack[17] grid. The cutoff energy in our calculation is 400 eV. All the calculations are relaxed to minimize the total energy of the system until the atomic forces are converged to 0.01 eV/Å. Period boundary conditions are applied in both in-plane and interlayer directions. Considering the exist of suspending bond, all our calculations are added the spin polarization. In the charge density calculation, we model a Pt particle with 19 atoms on the basal and edge plane of graphene (Fig. S9). We set the vacuum space larger than 15 Å to eliminate the interaction between periodic images of atoms. For the Pt particle on the edge plane, part of the boundary carbon atoms is passivated by hydrogen to equilibrate the electron[52].

## Data availability
The data supporting the findings of this study are available from the corresponding authors with a reasonable request.

## Code availability
Computer codes are available and freely downloadable from the LAMMPS (https://lammps.sandia.gov/index.html) and VASP (https://www.vasp.at/) websites.

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

## Acknowledgements

This project was not directly funded. We acknowledge the partial support from NSF-DMR award No. 1809439 and NSF-SNM award No. 1635221. This work made use of JOEL 3010 and JOEL JEM-ARM 200CF in the Electron Microscopy Core of UIC's Research Resources Center. JOEL JEM-ARM 200CF is supported by an MRI-R2 grant from the National Science Foundation DMR-0959470. Y. Liu acknowledges the use of the Center for Nanoscale Materials, an Office of Science user facility, supported by the U.S. Department of Energy, Office of Science, Office of Basic Energy Sciences, under contract no. DE-AC02-06CH11357. A portion of this work was conducted at Argonne National Laboratory. Argonne National Laboratory is operated for DOE Office of Science by UChicago Argonne, LLC, under contract no. DE-AC02-06CH11357. Z. Pang and T. Li acknowledge the University of Maryland supercomputing resources (http://hpcc.umd. edu) and Maryland Advanced Research Computing Center (MARCC) made available for conducting the research reported in this work.

## Author contributions

Z. Huang and R. Shahbazian-Yassar conceived the idea. Z. Huang designed the experiment with the direction of Y. Yao, L. Hu, and R. Shahbazian-Yassar. Z. Huang did the TEM experiments. K. He and X. Hu helped with the fast in situ TEM capture. Y. Yao and T. Li helped conduct the materials preparation. Z. Pang, J. Cheng, and T. Li performed DFT, MD and FEA simulations and analyzed the results. Y. Yuan, W. Yao, Y. Liu, S. Sharifi-Asl, K. Amine, and J. Lu provided necessary data analysis and discussion. Z. Huang, M. Cheng, and B. Song did the schematics of the paper. All the authors read the manuscript and assisted the revisions.

## Competing interests

The authors declare no competing interests.
