## [Peer Review File · Nature Communications]

Reviewers' comments:

Reviewer #1 (Remarks to the Author):

Review of the manuscript "Direct Observation of Metallic Nanoparticles Formation and Stabilization on Carbon Supports"

The manuscript presents in situ TEM observations of joule heated changes in carbon nanofibers and associated metal salts conversion into nanoparticles, as well as modelling of the nanoparticles stability.

Recommendation: Reject

The paper leads to fundamental questions on the methodology and conclusions that make me very cautious about trusting the conclusions.

1. Firstly, it is not clear how temperature is measured during the in situ experiment using the nanofactory holder. Without temperature characterized, how can conclusions be made?
2. Why isn't the Aduro TEM holder system used throughout since this apparently gives reasonably reliable temperature readings
3. A lot happens in 0.03 sec in Fig 2, and the CNF reorients in that time frame and still a clear image. The movies are not really showing what is going on – they are too short and too compressed and seem more to be two independent movies edited into one. We need the raw TEM image data in an full unedited version to be able to verify the conclusions
4. The CNF conversion should be better documented, e.g. they claim "These basal planes further come close to each other along Z direction³⁷" and this should be compared to observations.
5. Simulations seem to aim at showing the pinning of clusters on the edge planes. Comparison to the evolution on basal planes should be done as fig 5a show experimentally they should not adhere there. If the model cannot reproduce both features it is not a model of the system.
6. The models do not cover long enough times and do not study the essential aspects of the experiment and conclusions:
Movies from modelling show Pt atoms jumping to a new intercalation site across graphene edges in about 30 ps - that is quite fast compared to the 1 hr experimental time. Diffusion coefficient $D = \frac{1}{2} * MFP * V_{avg}$, with mean free path MFP 3Å, mean velocity $V_{avg} = 3Å/30ps$ we can estimate $D = 10^{-9} m^2/s$, which is large enough for diffusion to ripen the clusters making larger ones grow. Hence the modelling does not support the conclusion that clusters are stable, rather it does show there should be ripening.
7. The analysis of Fig 7 is cherry picking. Please perform a proper image analysis with statistically sound conclusions, rather than picking out a few clusters out of 100 to make these the foundation for any type of conclusion.
8. Diffusion/ripening of metal clusters on carbon is not new. The paper clearly lacks a relevant comparison to studies of diffusion on carbon to validate the results compared to other studies of such processes, instead of just referring to some recent papers on a joule heated synthesis process.

Reviewer #2 (Remarks to the Author):

The work "Direct Observation of Metallic Nanoparticles Formation and Stabilization on Carbon Supports" provided experimental evidence of high-temperature stable Pt nanoparticles supported on T-graphite. As pointed out by the authors, one should expect that the nanoparticles would fuse, but they observed good thermal stability of the clusters up to almost 1200 K upon annealing.

The authors provided enough detail on the experimental setup and the results they provided, from TEM and X-Ray, support most of their conclusions. In other words, I can see what they said they are seeing, the carbon support decorated by thermally stabilized nanoparticles. It is likely to be a problem with my computer, but I couldn't launch the movies provided as supplemental material. Their hypothesis on the mechanism that provided this stabilization makes sense, but then we arrive to the weakest point in the paper, in my opinion, which is the computational part.

I could not even understand the point of the DFT calculations. If it is energetically more favorable for the metallic atoms to bind to the graphite edges than to form clusters, and giving the large amount of those edges, should we not expect to see Pt-decorated edges rather than the nanoparticles? Maybe more interesting for the point of the paper would be an analysis of individual Pt diffusivity on the graphite edges and how would it cost, energetically, to break a Pt-Pt bond at the graphite edge.

Regarding the MD simulations, by analyzing Fig. 6, although the simulated time is too short, it seems that what the few simulation snapshots suggest is that the nanocluster is going to tear down with the atoms diffusing between graphite layers. In this sense, MD would be in line with the experimental observations on page 13, where the authors discuss that some nanoparticles disappeared. MD simulations at lower temperature would help to assess the thermal stability of the nanoparticles. Characterizing the nanoparticle crystal structure (it is clearly amorphous in the figures) is also necessary to shed light on this point.

Point-by-Point Response to Reviewers' Comments

Reviewer 1:

The paper leads to fundamental questions on the methodology and conclusions that make me very cautious about trusting the conclusions.

1. Firstly, it is not clear how temperature is measured during the in situ experiment using the nanofactory holder. Without temperature characterized, how can conclusions be made?

Reply to the Reviewer: We thank the reviewer for this instructive comment. The temperature of the sample during in situ experiment is an important parameter for this manuscript. In below, we provide new simulation results estimating the temperature during Joule heating, and also would like to clarify the reasoning behind our estimation for temperature based on the following:

1. We find that the carbon content in the carbon nanofiber changes significantly, from 82.7 wt% carbon to the final 97.1 wt% after Joule heating. Based on the literature (e.g. *Polymer Degradation and Stability* 2007, 92, 1421-1432; *Journal of Applied Polymer Science* 1991, 43, 589-600), achieving 96 wt% carbon concentration in the carbon nanofibers needs calcination temperatures of 1573K for one hour. While in our case, in milliseconds level, carbon content reaches to 97.1 wt%. Thus, we believe the temperature during this Joule heating is much higher than 1573K.
2. Another evidence (Figure R1) is based on our previous ex-situ Joule heating study (*Science*, 2018, 359 (6383), 1489-1494), that the formation of uniform nanoparticles was achieved by ~50ms joule heating process with the temperature measured to be ~2000K.

Therefore, we can conclude that during joule heating, the temperature is ultrahigh, larger than 1573K. However, for now, the direct measurement of temperature during the in situ TEM experiment is impossible considering the extremely small size of nanofibers and the current limitation of the Nanofactory technology for in-situ holders.

Figure R1. Temperature evolution measured during a ~ 50 ms ex-situ carbon-Joule heating process (reference: *Science*, 2018, 359 (6383), 1489-1494). (a) Spatial temperature evolution captured by a high-speed camera. (b) temporal evolution of temperature during the ~ 50 ms carbon Joule heating.

To visualize the temperature more clearly, we utilized finite element analysis (FEA) method about thermal analysis to reveal the temperature during the Joule heating process. The FEA method is widely accepted and used in researches for the simulation of evolution of temperature fields in many processes (e.g. *Nature* 2019, 569.7756: 388; *Nature communications* 2019, 10.1: 2067). In our FEA analysis, we first put a carbon nanofiber in contact with the gold electrode and the tungsten electrode, the same configuration as we did in the in situ TEM experiment. Then we applied the same input power ($40 \mu\text{W}$) on this setup. Thus, the temperature temporal evolution of the CNF is generated (Movie R1). The highest temperature was monitored to be 2079.4K (Figure R2a) at the center of the nanofiber within $16 \mu\text{s}$ and most of the CNF area is around 2000K within $10 \mu\text{s}$ (Figure R2b). This ultrafast temperature evolution confirms our estimation that the temperature during our Joule heating experiment is higher than 1573 K.

Figure R2. Finite element analysis of the Joule heating process upon the applying of a 40 μW input power. (a) Temperature profile of the unit at 10 μs when the system has reached a thermal equilibrium. For clarity, a quarter of the model is removed from the result and the electrodes are partially shown. (b) Temporal evolution of the temperature at the monitor point (shown in Figure R2a) inside the CNF. Most of the center part of the CNF reaches ~ 2000 K at around 10 μs .

Revision in the manuscript: We add Figure R2 as Figure 4 in the manuscript with the following description on Page 9. And added the Movie R1 as Movie S5 in supplementary information.

“The temperature evolution on the CNF during Joule heating are studied through finite element analysis (FEA). As demonstrated in Figure 4a, a CNF is bridged between the Au and W electrodes. After applying the input power of 40 μW (used in the in situ experiment), the center areas of the CNF reached to a thermal equilibrium at around 2000 K. More precisely, the temporal evolution of temperature at the center point of CNF is shown in Figure 4b. The temperature rapidly raises to ~ 1600 K within 4 μs and reaches to an equilibrium of ~ 2000 K at 10 μs . Based on the Joule’s first law, heat conduction always exists. Thus, the areas close to the electrodes show a temperature gradient towards lower temperature. The temperature evolution of the entire CNF is visualized in Movie S5.”

2. Why isn’t the Aduro TEM holder system used throughout since this apparently gives reasonably reliable temperature readings

Reply to the Reviewer: We thank reviewer for this suggestion. We believe the extreme fast heating rate inducing by Joule heating is key to many findings of this manuscript (e.g. graphitization of amorphous carbon, stabilization of nanoparticles on graphitic edges, homogeneous chemical mixing of multielemental nanoparticles, uniform distribution of metallic nanoparticles, etc). We performed some tests with Aduro-heating system (see below results) but these key findings are not observed. We believe, studies of heating rates and their impacts on kinetics/structural evolution of nanoparticles and carbon fibers deserve a thorough investigation in a separate paper with a different focus.

Here, we have outlined more details about such differences and also some results of in situ heating with Aduro heating holder:

- (A) In the Aduro TEM holder, the heating rate is several orders of magnitude lower (hundreds of Kelvin per second) than the Joule heating process. In our Joule Heating method, the heating speed is larger than 10^5 K/S (as confirmed by the temperature simulation Movie R1 and in situ videos R2 & R3). This high heating rate is essential

to alleviate the diffusion and agglomeration of metal ions/particles. Thus, the Aduro TEM heating system could not meet our demand.

(B) To alleviate the reviewer's concern, we performed in situ heating by using the Aduro TEM holder at the maximum heating rate of 200 K/s (Figure R3). No expansion and graphitization of amorphous carbon substrate was observed. The Pt nanoparticles appear at temperatures between 473 to 673K (Figure R3b) but they continually grow to larger sizes (Figure R3c-3f) over time. Due to slow rate of process, it resembles salt dehydration/decomposition and metal nanoparticles formation (Movie R2).

Figure R3. In situ heating through the Aduro heating holder at a heating rate of 200 K/s. (a)-(f), Pt nanoparticles nucleation and growth on CNF, red arrows indicate areas with clear particle nucleation and growth.

(C) The relatively slow heating process in Aduro heating system yield in inhomogeneous mixing of metallic elements. As we can see from Figure R4a, the EDS mappings show the three element nanoparticles synthesized through Aduro heating system (heating speed of 200 K/s) with chemical segregation. While in the CNF-Joule heating system, all three elements are well mixed (Figure 5f). In Joule heated CNFs, the temperature of carbon fibers raise to extreme high temperatures in μs time span leading to an almost simultaneous decomposition and nucleation of all three salts/elements, enabling well mixing or alloying of all elements at the single nanoparticle level. While in Aduro heating system, due to the relatively slowing heating, the salts decompose one by one leading to a poor mixing of nanoparticles.

(D) In addition, The CNF substrate evolution is different: In CNF Joule heating, the CNF substrate expands and crystallizes (Figure 2) due to CNF itself generates high temperatures when power applies. In Aduro TEM holder, the heating source is the MEMS chip and CNF doesn't change quite much under the radiation heating from MEMS chip (from Figure R3a to R3f). The CNF substrate is also studied under high resolution TEM. As shown in Figure R4b, the substrate is still quite smooth compared to the ones after CNF-Joule heating (Figure 5a).

Figure R4. Multicomponent nanoparticles synthesized through Aduro heating system with a heating rate of 200 K/s. (a) EDS mappings show the elemental distribution of Pt, Pd and Ni. (b) high resolution TEM image of smooth CNF surface after the formation of PtPdNi nanoparticles. Scale bar is 5 nm.

Revision in manuscript: We add the above results in the manuscript and revised the manuscript on Page 7:

“To compare with other heating methods, we performed in situ heating experiments on S-CNFs utilizing microfabricated-heating SiN membrane systems. The major differences between the two heating platforms are the significantly slower heating rate (heating speed $\sim 200\text{K/s}$) in the membrane devices versus the Joule heating ($>10^5\text{ K/s}$). No graphitization or expansion of amorphous carbon substrate was observed. The Pt nanoparticles appear at temperatures between 473 to 673K (Figure S4b) but they continually grow to larger sizes (Figure S4c to S4f) over time. Due to slow rate of process, it resembles metal salt dehydration/decomposition and metal nanoparticles formation (Movie S5). Due to slower rate of heating rate and lack of graphitization, it was not good to obtain multielement alloyed nanoparticles without chemical segregation (Figure S4g).”

Figure R3 and R4 has combined to Figure S4 and put into the supporting information. Movie R2 has added in the supporting information as Movie S5.

Figure S4. In situ heating with SiN-based device TEM holders at heating rate of 200K/s. (a)-(f) The formation of Pt nanoparticles on S-CNF marked by the red arrows ; (g) EDS maps show the elemental segregation of Pt, Pd and Ni after heating; and (h) high resolution TEM image the smoothness of CNFs surface after the formation of PtPdNi nanoparticles. Scale bar 5nm.

3. A lot happens in 0.03 sec in Fig 2, and the CNF reorients in that time frame and still a clear image. The movies are not really showing what is going on – they are too short and too compressed and seem more to be two independent movies edited into one. We need the raw TEM image data in a full unedited version to be able to verify the conclusions

Reply to the Reviewer: We thank reviewer for the comment. We want to point out that, the only reason for this distinct change is because this process is ultrafast that is beyond the recoding limit of our CCD camera (30 frames per second(fps)).

To address the reviewer's concern, we collaborated with the electron microscopy group at Northwestern University (NU) to perform the same experiments in their advanced microscope equipped with ultrafast CMOS camera (capable of ≥ 200 fps, achieving less than 5ms time resolution). We repeated the same joule heating process and as can be seen in our new data, this Joule heating process is consistent and ultrafast. Figure R5 (corresponding to

Movie R3, 300 fps) shows an example of this Joule heating process that takes less than 7ms. The salt loaded CNF (S-CNF) is shown in Figure R5a right before the Joule heating. By the application of only 40 μ W power, the CNF expansion and nanoparticles formation are captured after 3.33 ms as shown in Figure R5b. Due to ultrafast nature of the entire process, Figure R5b is in fact an overlap of the original S-CNF image (Figure R5a) and the final CNF expansion and nanoparticles formation image (Figure R5c). This means that the entire process is faster than 3.33 ms. Figure R5d is imaged at the end of this process, which shows clear wrinkled structure and nanoparticles on CNF. The left portion of the CNF is burned, which should be due to point contact between tungsten and CNF (marked as red arrow in Figure R5a) leading to the passage of current from a small area (high current density). It generates extremely high temperature at the contact area during Joule heating that burns out the CNF.

Figure R5. In situ Joule heating of a salt loaded CNF (S-CNF) captured by ultrafast camera (300 fps) at low magnification. (a) Pristine status of S-CNF before the start of Joule heating. (b) TEM image taken at after 3.33 ms, where S-CNF expansion and the formation of Pt nanoparticles happen. The pristine S-CNF shadow is still existing in the background of the image due to camera recording being at the image capturing limit. (c) The TEM image of nanoparticles on expanded CNF captured only after 6.67 ms. (d) higher resolution TEM image of the Joule heated S-CNF with wrinkled structure and nanoparticles.

Another video (Movie R4, 200 fps) is recorded at higher magnification to focus on a small region of the CNF edge to show the details of the transition during Joule heating. As shown in Figure R6, a similar flash evolution is captured. Figure R6a represents the pristine status of the S-CNF, which is located at the upper left part, the lower right is the vacuum area (marked by red arrows). The dark spots and layer on the CNF surface (marked by yellow arrows) is the loaded Pt salt crystals. The structural evolution of S-CNF already happens by 5 ms of Joule heating, as shown in Figure R6b. In Figure R6b, a similar ghost image which

contain the original fiber image together with the features at 10 ms (Figure R6c), again confirms that the total time evolution of this process is less than 5 ms. The process is associated with the CNF expansion and nanoparticles formation mostly on the wrinkled structures.

Figure R6. In situ Joule heating of S-CNF captured by ultrafast camera (200 fps) at high magnification. The process is associated with the CNF expansion and nanoparticles formation mostly on the wrinkled structures.

Revision in the manuscript: We have added Figure R5 and Figure R6 as Figure S1 and S2 in the Supplementary information and included the following sentences in the revised manuscript Page 5. Movie R3 and R4 has also been added as Movie S3 and S4.

“In situ experiments are also repeated and captured by ultrafast CMOS camera at an image acquisition speed of ≥ 200 fps achieving less than 5 ms time resolution (Movie S3 and S4). The evolution is still the same, as shown in Figure S1 and Figure S2 which means that the nanofiber structural evolution and nanoparticles formation happens within 5 ms.”

4. The CNF conversion should be better documented, e.g. they claim “These basal planes further come close to each other along Z direction³⁷” and this should be compared to observations.

Reply to the Reviewer: We thank the reviewer for the comments. The modeling work referenced here has given some insights on the intermediate process of the amorphous carbon nanofiber graphitization transition [e.g. *CRYSTALLOGRAPHY REPORTS C/C OF KRISTALLO- GRAFIJA 1999, 44, 749-754*]. While it is expected that by graphitization the basal planes stack on top of each other, it is beyond our microscope temporal and spatial resolution to observe such phenomenon. Per the reviewer’s comment, we modified our hypothesis to be based on the original amorphous carbon structure changes to the final graphitized structure per our observation (Figure R7).

Figure R7. The schematic depicts the amorphous CNF to T-graphite transformation at high temperatures.

Revision in the manuscript: We added Figure R7 to the Figure 3 as a new Figure 3e. We also revised the sentence in Page 8.

“The amorphous CNF are not fully transformed to crystalline graphite but an intermediate T-graphite structure.”

Figure 3. TEM and EDS analyses of pristine amorphous CNF evolution during Joule heating. (a) TEM image of original CNF without salt loading where the corresponding SAED pattern displays diffusive rings characteristic, representative of an amorphous structure. Scale bar is 50 nm. (b) TEM image of wrinkled CNF achieved after Joule heating, where the SAED pattern shows sharp rings, corresponding to T-graphite. Scale bar is 50 nm. (c) HRTEM image of the wrinkled structure and (d) quantitative EDS analysis of carbon, nitrogen and oxygen content in the CNF before and after Joule heating. (e) The schematic depicts the amorphous CNF to T-graphite transformation at high temperatures.

5. Simulations seem to aim at showing the pinning of clusters on the edge planes. Comparison to the evolution on basal planes should be done as fig 5a show experimentally they should not adhere there. If the model cannot reproduce both features it is not a model of the system.

Reply to the Reviewer: We thank Reviewer #1 for the constructive comments.

In the MD simulations, we cannot simulate the entire time span of experimental process due to computational time expense constraint. Therefore, the simulations focus on revealing

the underlying mechanism of the experimental results. Considering the limitation of simulation, we have to make certain assumptions to fit better with the experimental conditions. In the simulations, all atoms are located in the vacuum environment with no gravity. The random arrangement of atoms can lead part of them to go everywhere. Thus, we assume that the atoms have formed the cluster and pin on the edge and basal planes. Then we calculate the interaction between cluster and different planes to analyze the derivation.

To verify our analysis and follow the reviewer's suggestion, we also show the derivation of the cluster on the basal plane (Figure R8a). It can be found that the cluster moves around and cannot pin on the plane stably. However, the cluster is pinned tightly on the edge plane. Although a few atoms intercalate the graphene layers (Figure R8b). It means that clusters are much more stable on the edge plane.

Figure R8. The morphological derivation of Pt cluster in the presence of T-graphene with (a) basal planes and (b) edge plane at 1800K.

Revision in manuscript: We have added Figure R5a to the supporting information as Figure S6 and revised manuscript on Page11:

“When the Pt cluster adheres to the basal plane of T-graphite (Figure S6), it can be found that the cluster moves around and cannot pin on the plane stably.”

6. The models do not cover long enough times and do not study the essential aspects of the experiment and conclusions: Movies from modelling show Pt atoms jumping to a new intercalation site across graphene edges in about 30 ps - that is quite fast compared to the 1 hr experimental time. Diffusion coefficient $D = \frac{1}{2} * MFP * V_{avg}$, with mean free path MFP 3\AA , mean velocity $V_{avg} 3\text{\AA}/30\text{ps}$ we can estimate $D = 10^{-9} \text{ m}^2/\text{s}$, which is large enough for diffusion to ripen the clusters making larger ones grow. Hence the modelling does not support the conclusion that clusters are stable, rather it does show there should be ripening.

Reply to the Reviewer: We thank Reviewer #1 for the constructive comments.

We aim to use MD simulations to find the micro phenomenon through statistical analysis after equilibrium, which is not possible to observe directly by experiments. Following the reviewer's suggestion, we also extend the simulation time (Figure R9a). The cluster maintains the thermal oscillation after verging to equilibrium ($t \sim 30\text{ps}$). More relaxation time doesn't influence the final structure and our analysis. Figure R9b also shows the variation of total energy as the function of relaxation time. It is verified directly that the cluster converges

to the stable structure after $t=30\text{ps}$. In our manuscript, we pick the period with 60ps to be long enough to analyze the mechanism. Meanwhile, it can be found that the same time scale (ps) is widely used in the other simulations about the interaction between cluster and graphene {*science* 2014, 343, 6172, 752-754; *The Journal of Physical Chemistry C* 2012, 116 (21), 11776-1178}.

The reviewer mentions that the Pt jumps into intercalation site $\sim 30\text{ps}$, and raise assumption that given long enough time, the Pt cluster will diffuse and ripen together. That is not true. Actually, at the first 30ps, the system is at nonequilibrium state, as we can see from the total energy curve (Figure 9b), large energy fluctuation happens. That's why the Pt atoms has such high diffusion coefficient. However, after $\sim 30\text{ps}$, the system reaches to an equilibrium status. Where the total energy is stable, which also means the motion of Pt atoms is limited. In this case, it does not carry much value to study the instantaneous motion of atoms at nonequilibrium status and then shed light on the entire heating process.

Figure R9. (a) The morphological derivation of Pt cluster on the edge plane of T-graphene. (b) Total energy as the function of relaxation time.

Revision in manuscript: We have put Figure R9 as Figure S10 in the supporting information and added the following text in the manuscript and cited the literature reports (Page 16):

“for up to 120ps (Figure S10), which is long enough to analyze the mechanism⁵¹⁻⁵².”

7. The analysis of Fig 7 is cherry picking. Please perform a proper image analysis with statistically sound conclusions, rather than picking out a few clusters out of 100 to make these the foundation for any type of conclusion.

Reply to the Reviewer: Per the reviewer’s comments, we analyzed the particles at room temperature (Figure R10a) and 1173K (Figure R10b) as shown in Figure R10c & d, and overlapped in Figure R10e. The detailed statistics analysis is shown in tables R1 and R2: (1) about 78% of the particles barely move, and still largely overlap after 1173K treatment. (2) About 16% of the particles were disappeared. Two locations where the particles disappear are discussed in the manuscript (Figure 10e&f). There is only about 4% of new particles formation (may be due to Ostwald ripening) and 3% of particles moves (may not adhere to the edge plane quite well). Moreover, the average size of the nanoparticles does not change (3.2 nm vs 3.2 nm). Thus, we can safely conclude that the Joule heating formed nanoparticles are ultra-stable at elevated temperatures. The total area of the nanoparticles reduced $\sim 12.4\%$

(from 280.5 to 245.6 nm²), which is the result as we observed and discussed in the manuscript: nanoparticles change to high density single atoms and bond on carbon substrate. Similar particle to single atom result has also been reported recently (*Nature nanotechnology* 2018, 13.9: 856), in which the researchers find that noble metal nanoparticles could convert to single atoms and dispersed on substrate above 1173K, totally in line with our observations.

Figure R10. Image analysis on the particles at RT and 1173K. (a) The HAADF image of the nanoparticles at RT. (b) The green lines show the border of nanoparticles at RT. (c) The HAADF image of the nanoparticles after 1173K treatment. (d) The purple lines show the border of nanoparticles at 1173K. (e) The overlapped image of the particle to illustrate the movement of nanoparticles by heating from RT to 1173K.

Table R1. The relative motion of particles by comparing the RT and 1173 K treatments

The status of nanoparticles motion	Percentage
Fixed particles (overlap)	78%
Disappeared particles	16%
Newly appeared particles	4%
Moved particles (not overlap)	3%

Table R2. Area and average size of the nanoparticles

Sample	Particle area (nm ²)	Average particle size (nm)
RT	280.5	3.2
1173 K	245.6	3.2

Revision in the manuscript: We have added Figure R10, Table R1 and R2 in supporting information as Figure S9 and Table S1 and S2; added the literature of particle convert to single atom report in the manuscript. And revised the manuscript in Page 12-13:

“To clearly review the changes happen at 1173 K, a thorough analysis of all the particles is performed by comparing the TEM images taken at RT and at 1173 K. As shown in

Figure S9, all the outlines of the particles were marked (Figure S9b,d) and overlapped (Figure S9e). The detailed statistics analysis is shown in table S1 and S2: (1) About 78% of the particles barely move, still largely overlap with the RT locations after 1173K treatment. (2) About 16% of the particles were disappeared. (3) Only about 4% of new particles appeared (may due to Ostwald ripening) and 3% of particles moved (possibly due to lose adhering to the edge planes). (4) The average size does not change (3.2 nm vs 3.2 nm), while the total area of the nanoparticles reduced 12.4% (from 280.5 to 245.6 nm²).

“Very similar particle transform to single atom phenomenon has also been reported recently⁴⁶. At 1173K and above, the noble metal nanoparticles could be converted to single atoms and absorbed on carbon substrate.”

8. Diffusion/ripening of metal clusters on carbon is not new. The paper clearly lacks a relevant comparison to studies of diffusion on carbon to validate the results compared to other studies of such processes, instead of just referring to some recent papers on a joule heated synthesis process.

Response: We thank the reviewer for the insight. Citing and comparing to similar works are important to represent the advantage and importance of our study. First of all, we need to mention that the reports of ultrafast Joule heating synthesis of nanoparticles on carbon support are very new (since 2017-2018). That is why the literature citing Joule heating-induced synthesis of nanoparticles are relatively recent. This methodology is unique on several points, which there is hardly any other methods could catch up: (1) ultrafast (tens of milliseconds based on recent Joule heating synthesis works); (2) ultrahigh temperature during this process (~2000K), as measured and state in the referenced Joule heating works and our FEA modeling in Figure R1; (3) ultrafast substrate phase transition, amorphous carbon crystallize within 5 ms based on our recent captured movies R1&R2.

Per the reviewer’s suggestions, we compared our work with the reports sharing similar carbon substrate feature and metal clusters/nanoparticles. In an in situ TEM heating studies on the stability of Pt nanoparticles on graphene (*Nano Research* 2011, 4, (5), 511-521), the Pt nanoparticles happened to merge/ripen at 703K (430 °C). Another work (*The Journal of Physical Chemistry C* 2012, 116, 16, 9274-9282) also shows above 673 K (300 °C), the Pt nanoparticles on graphene migrate and agglomerate to larger sizes. While in our study, we found out that the Pt nanoparticles were stable till 1173K. We also noticed that at 1173K, nanoparticles slowly concerted to single atoms. Very similar phenomenon has also been reported recently (*Nature Nanotechnology* 2018, 13, (9), 856): Above 1173K, the noble metal nanoparticles could be converted to single atoms and absorbed on carbon substrate. However, in their work, the nanoparticles were not stable on carbon substrate at all. The diameter of the particles increased by heating from 373 K to 1173K all the time.

As a conclusion, the works with similar setup or phenomenon about heating nanoparticles on carbon supports show that the nanoparticles are not stable at low temperatures (< 673 K). In our work, the particles are stable till 973K, and slowly concerted to single atoms at 1173 K (the location of particles were fixed with limited movement). Since

the particles synthesized through Joule heating improves the stability by more than 300 K, we can safely conclude that the nanoparticles are ultra-stable through Joule heating approach.

More works about stable metal on carbon substrates were done by simulations. And these works mainly focused on the interaction between metallic nanoparticles and graphitic basal planes. {e.g. *ACS nano* 2010, 4, 8, 4920-4928; *Journal of the American Chemical Society* 2012, 134, 7, 3472-3479}. There may exist various defects on basal planes which influence the adsorption of nanoparticles {e.g. *The Journal of Physical Chemistry C* 2012, 116, 11, 6543-6555; *Computational Materials Science* 2015, 96, 268-276; *The journal of physical chemistry letters* 2012, 4, 1, 147-160}, while the temperature effect is ignored in their simulations. Thus, there is a lack of understanding about the stability of nanoparticles on carbon substrates, especially on edge planes, at elevated temperatures.

Revision in the manuscript: We have added these literature reports in the manuscript and revised manuscript on Page 3:

“A number of simulations focus on the interaction between the metallic nanoparticles and graphitic basal planes^{23,24} or defective basal planes²⁵⁻²⁷. While the temperature effect is ignored in their simulations. Thus, there is a lack of understanding about the stability of nanoparticles on carbon substrates at elevated temperatures.”

And Page 13:

“Some other high temperature works of noble metals on carbon based substrates show the metal nanoparticles are not stable on substrate at less than 673 K⁴⁶⁻⁴⁸. Since the particles synthesized through Joule heating improves more than 300 K stability, we can safely conclude that the Joule heated particles are ultra-stable.”

Reviewer #2:

The work "Direct Observation of Metallic Nanoparticles Formation and Stabilization on Carbon Supports" provided experimental evidence of high-temperature stable Pt nanoparticles supported on T-graphite. As pointed out by the authors, one should expect that the nanoparticles would fuse, but they observed good thermal stability of the clusters up to almost 1200 K upon annealing.

The authors provided enough detail on the experimental setup and the results they provided, from TEM and X-Ray, support most of their conclusions. In other words, I can see what they said they are seeing, the carbon support decorated by thermally stabilized nanoparticles. It is likely to be a problem with my computer, but I couldn't launch the movies provided as supplemental material. Their hypothesis on the mechanism that provided this stabilization makes sense,

but then we arrive to the weakest point in the paper, in my opinion, which is the computational part. I could not even understand the point of the DFT calculations. If it is energetically more favorable for the metallic atoms to bind to the graphite edges than to form

clusters, and giving the large amount of those edges, should we not expect to see Pt-decorated edges rather than the nanoparticles? Maybe more interesting for the point of the paper would be an analysis of individual Pt diffusivity on the graphite edges and how would it cost, energetically, to break a Pt-Pt bond at the graphite edge.

Reply to the Reviewer: We thank Reviewer #2 for the constructive comments.

We are using DFT calculations to show the electronic interaction between metallic atoms and graphite without temperature interference. The binding energy of Pt and carbon (-8.51eV/atom) is significantly higher than the cohesive energy of Pt cluster (-5.37eV/atom), which means Pt prefers to distribute in the areas containing carbon dangling bonds, instead of forming larger nanoparticles. And there exists a large amount of dangling bond on the edge plane. As shown in literatures {*ACS Catalysis* 2016, 6, 2642-2653; *Nature* 2007, 446, 60} and Figure 3 discussions in our manuscript. Meanwhile, our DFT calculations also show the accurate energy (5.37eV) to break the Pt-Pt bond. Due to the larger binding energy of Pt-C, we do find a few Pt atoms diffuse on the edge plane in our MD simulation (Figure 6 and Figure R9a), forming the Pt decorated edge. But the diffusion reaches saturation quickly, the total energy is stable after 30 ps (Figure R9b). Which means Pt atoms diffuse on edge plane is limited after reach to equilibrium status. Then, the cluster is pinned on the edge plane stably.

Per reviewer's comments, we also simulate the diffusivity of single atom on the edge plane at high temperature (T=1800K), shown as Figure R11. It can be concluded that the binding of the Pt and carbon atoms is strong enough to resist the moving. The position of Pt atom relative to the edge plane doesn't change from 0 ps to 120 ps.

Figure R11. *The morphological derivation of individual Pt atom in the presence of T-graphene with basal planes*

Revision in the manuscript: We have added Figure R11 as Figure S5 in the supporting information and added the following sentence in Page 11:

“The intercalated atom is relatively stable on the edge plane by our MD simulations (Figure S5).”

Regarding the MD simulations, by analyzing Fig. 6, although the simulated time is too short, it seems that what the few simulation snapshots suggest is that the nanocluster is going to tear down with the atoms diffusing between graphite layers. In this sense, MD would be in line with the experimental observations on page 13, where the authors discuss that some nanoparticles disappeared. MD simulations at lower temperature would help to assess the

thermal stability of the nanoparticles. Characterizing the nanoparticle crystal structure (it is clearly amorphous in the figures) is also necessary to shed light on this point.

Reply to the Reviewer: We thank Reviewer #2 for the constructive comments.

Following the reviewer's comment that the simulation time is too short, we double the simulation time, as shown in Figure R9b. After a few atoms intercalated, the whole structure will be stable. The cluster maintains the thermal oscillation after verging to equilibrium ($t \sim 30$ ps). More relaxation time doesn't influence the final structure and our analysis. Figure R9b also shows the variation of total energy as the function of relaxation time. It is verified directly that the cluster converges to the stable structure after $t = 30$ ps. In our manuscript, we pick the period with 60ps is long enough to analyze the mechanism. Meanwhile, it can be found that the same time scale (ps) is widely used in the other simulations about the interaction between cluster and graphene {e.g. *science* 2014, 343, 6172, 752-754; *The Journal of Physical Chemistry C* 2012, 116 (21), 11776-1178}.

In our MD simulation, parts of atoms intercalate into graphene layers and bond on the carbon substrate. It may result in the disappearance of nanoparticles. Thus, we agree with reviewers' comment. It may more likely that the nanoparticles convert to single atoms and disperse on the substrate, as we observed in Figure 7f-g. Very similar particle transform to single atom phenomenon has also been reported recently {*Nature Nanotechnology* 2018, 13(9), 856}. At 1173K and above, the noble metal nanoparticles could be converted to single atoms and absorbed on carbon substrate.

We also calculate the thermal stability of the nanoparticle at a lower temperature ($T = 1000$ K), as shown in Figure R12a. There also exist a few atoms intercalate into the graphite layers and the nanoparticle is stable on edge plane based on the total energy (Figure R12b).

Figure R12. (a) The morphological derivation of amorphous Pt cluster in the presence of T-graphene with basal planes at 1000K. (b) Total energy as the function of relaxation time.

Following the reviewer's suggestion, we also calculate the derivation of the nanoparticle crystal structure (Figure R13). At the high temperature ($T = 1800$ K), the crystal structure is turned into the amorphous structure and the derivation is the same with amorphous one (Figure 6 and Figure R8b).

Figure R13. The morphological derivation of the crystalline Pt cluster in the presence of T-graphene with edge planes at T=1800K.

Revision in the manuscript: We added Figure R12 as Figure S7 in the supporting information, and added the above analyses on Page 11:

“Lower temperature (1000K) condition is also simulated (Figure S7), which shows similar phenomenon.”

And Page 13:

“Based on our MD simulation, parts of atoms intercalate into graphene layers and bond on the carbon substrate. It may result in the disappearance of nanoparticles. It may more likely that the nanoparticles convert to single atoms and disperse on the substrate, as discussed here. Very similar particle to single atom conversion has also been reported recently⁴⁶ in which the researchers find that noble metal nanoparticles could convert to single atoms and dispersed on substrate above 1173K, totally in line with our observations.”

Reviewers' comments:

Reviewer #1 (Remarks to the Author):

The authors have provided a thorough reply, which however still misses some of the points I raised, and leads to new questions, that I would consider important to get right if this is to be published:

1 – In your new additions, clearly the diffusion is far more present on the basal plane graphene in fig R8 in reply to my question 5, so relatively speaking you make a case. But the diffusion away from the particle on the edge plane is still present and quickly dismissed rather than discussed. Diffusion is a random walk and some Pt will return to the particle while others will end up away from it – and that will take time. I find this key point raised earlier missing in your paper and should be addressed appropriately as it still leads to the question if you can conclude anything from the simulations.

2- One atom leaving the cluster would change the energy of the cluster with what seems to be a few eV and hence not at all possible to determine on the scale of your energy vs time plot with a unit of 10^4 eV. Hence your energy plot cannot be used to show diffusion is absent as you would like to conclude from it.

3 – interestingly your Pt on edge planes stay fixed in simulation R11 while they seem to be jumping on and off and around the clusters in the particle simulations. Doing statistics on a single atom for a very short time in a simulation is not a foundation for discussing diffusion. You will need to study an ensemble or time series over different locations covering enough scenarios of bonds etc to actually assess the diffusion.

4- How can your plot of energy vs time of relaxation of the cluster show an increasing energy from 5 to 120 ps? Shouldn't it be relaxing and lowering energy? is this an increase due to atoms leaving the cluster?

5 – From your discussion of differences between the two heater systems, there should also be a discussion if thermo or electromigration could influence the process, as your observations are done under presumably high of several microamperes in a nanoscale fiber (though still it may be on too short time scale for this to be effective).

6 the recent paper in nature nanotech 13 p 856-861 appears to provide a detailed description of the presumed cause of the stability. In your paper you merely state that it is stable, while still implying single Pt stability. What makes the Pt stable in those positions as in R11 – what functional groups are included in your model and what bonds are formed?

7 what terminating functional groups are used in the simulation and what would be relevant to use? These could considerably influence the systems, e.g. -H vs -OH on the edge planes.

8 - In your long time stability test, it is not clear how the ex situ formed sample was formed – which should be clarified, and also lead on to the question how you foresee this synthesis method can be scaled up to any reasonable amount of relevance? Unless of course you provide more detailed chemical insight to the underlying cause of the stability in the above questions that chemists then can try to replicate.

Reviewer #2 (Remarks to the Author):

On page 261 -> "density functional theory (DFT).

The authors improved the simulations and the discussion of the results that they presented in the previous version of the paper. The discussion about the pinning mechanisms could be more polished though.

What I guess, from experimental and simulational results: larger particles (how big should they be?) are anchored to graphene edges through Pt atoms that intercalated into the planes. We should therefore have three distinct regions to consider: one in which we basically have graphene with a few, isolated Pt atoms intercalated, far from the edges; a second one quite rich in Pt, forming a stable second phase (a graphene-Pt alloy?), near the edges; and the Pt particle itself, attached to the Pt-rich edges through Pt-Pt and Pt-C bonds strong enough to impede particle glide. On the other hand, the small particles, the ones that eventually disappeared, were too small and got dissolved by continuously losing atoms that diffused into the graphene planes.

In any case, full elucidation of these mechanisms would require larger and longer simulations and systems of different sizes.

Point-by-Point Response to Reviewers' Comments

Reviewer 1:

The authors have provided a thorough reply, which however still misses some of the points I raised, and leads to new questions, that I would consider important to get right if this is to be published:

1 – In your new additions, clearly the diffusion is far more present on the basal plane graphene in fig R8 in reply to my question 5, so relatively speaking you make a case. But the diffusion away from the particle on the edge plane is still present and quickly dismissed rather than discussed. Diffusion is a random walk and some Pt will return to the particle while others will end up away from it – and that will take time. I find this key point raised earlier missing in your paper and should be addressed appropriately as it still leads to the question if you can conclude anything from the simulations.

Reply to the Reviewer: We thank the reviewer for this constructive comment. To address this comment, we first calculated the potential energy of a Pt particle on the basal plane and edge plane of graphene, as shown in Figure R1. It can be found that the potential energy of the Pt particle on the basal plane keeps decreasing gradually, which means the particle is not stable. By contrast, the potential energy of the Pt particle on the edge plane decreases drastically and finally reaches a plateau value significantly lower than the potential energy of the Pt particle on the basal plane. Such a comparison demonstrates that the Pt particle prefers to pin on the edge plane, rather than the basal plane.

Figure R1 Potential energy of the Pt particle as the function of relaxation time in the presence of T-graphene with the basal plane and the edge plane.

To further demonstrate the stability on the edge plane of graphene, we also calculated the charge density by DFT method. Considering the available computational resource, we model a Pt particle with 19 atoms on the basal and edge plane of graphene (Figure R2). We set the vacuum space larger than 15 Å to eliminate the interaction between periodic images of atoms. For the Pt particle on the edge plane, part of the boundary carbon atoms are passivated by hydrogen to equilibrate the electron [*The Journal of Physical Chemistry C*, 117 (2013) 25424-25432]. Figure R2b shows the charge density difference of Pt on the basal plane. There exists few charge transfer between

the Pt particle and the graphene plane, which suggests that the main interaction between the Pt particle and the graphene basal plane is van der Waals force and it is hard to form the covalent bond between these two atoms. On the contrary, large amount of charge transfer occurs between the Pt particle and the graphene edge plane (Figure R2d), which suggests strong binding energy between the Pt particle and the edge plane of graphene (in agreement with MD simulation results in Figure R1).

The above two types of simulations clearly show that the diffusion of the Pt particle on the edge plane is much harder than that on the basal plane.

Figure R2 (a) Atomistic structure of the Pt particle on the basal plane and (b) the corresponding charge density. (c) Atomistic structure of the Pt particle on the edge plane and (d) the corresponding charge density. Brown, silver and pink balls represent the C, Pt and H atoms, respectively. Yellow and blue regions represent the gain and loss of electrons and the scale level is set as $\pm 0.015e$.

Revision in the manuscript:

Figure R1 and R2 have been added in the supporting information as Figure S8 and S9. The following has been added to the revised manuscript on page 12:

“Figure S8 shows the potential energy of the Pt particle as the function of relaxation time in the presence of T-graphene with basal plane and edge plane. The potential energy of the Pt particle on the basal plane keeps decreasing gradually, which means the particle is not stable. To further demonstrate the stability on the edge plane of graphene, we also calculated the charge density by DFT method. Figure S9b shows the charge density difference of Pt on the basal plane. There exists few charge transfer between the Pt particle and the graphene plane, which suggests that the main interaction between the Pt particle and the graphene basal plane is van der Waals force and it is hard to form the covalent bond between these two atoms. On the contrary, large amount of charge transfer occurs between the Pt particle and the graphene edge plane (Figure S9d), which

suggests strong binding energy between the Pt particle and the edge plane of graphene (in agreement with MD simulation results in Figure S8). The above two types of simulations clearly show that the diffusion of the Pt particle on the edge plane is much harder than that on the basal plane.”

The following has been added to the revised manuscript on page 19:

“In the charge density calculation, we model a Pt particle with 19 atoms on the basal and edge plane of graphene (Figure S9). We set the vacuum space larger than 15 Å to eliminate the interaction between periodic images of atoms. For the Pt particle on the edge plane, part of the boundary carbon atoms is passivated by hydrogen to equilibrate the electron [*The Journal of Physical Chemistry C*, 117 (2013) 25424-25432].”

2- One atom leaving the cluster would change the energy of the cluster with what seems to be a few eV and hence not at all possible to determine on the scale of your energy vs time plot with a unit of 10^4 eV. Hence your energy plot cannot be used to show diffusion is absent as you would like to conclude from it.

Reply to the Reviewer: We thank the reviewer for this constructive comment. We calculated the potential energy curve of the Pt particle as the function of relaxation time. The scale of the energy is suitable now, and we can conclude that the particle on edge plane becomes stable after 60ps, as shown in Figure R1 (Red curve).

3 – interestingly your Pt on edge planes stay fixed in simulation R11 while they seem to be jumping on and off and around the clusters in the particle simulations. Doing statistics on a single atom for a very short time in a simulation is not a foundation for discussing diffusion. You will need to study an ensemble or time series over different locations covering enough scenarios of bonds etc to actually assess the diffusion.

Reply to the Reviewer: We thank the reviewer for this constructive comment. In the MD simulation, all the atoms are in a dynamic process. Due to the fluctuation of carbon atoms, it looks like the Pt atom is jumping on and off. Actually, the relative location of Pt and graphene is constant. It can also be found in Figure S6.

Figure S6. The morphological derivation of individual Pt atom in the presence of T-graphene with edge planes.

We also consider the bond configuration of Pt-carbon based on the X-ray absorption

spectroscopy (XAS) data:

Previously, we studied the Pt bond formation with the same carbon substrate. We further diluted the Pt concentration toward single atom dispersion to form Pt-X bonds with the carbon substrate. Based on the Extended X-ray Absorption Fine Structure (EXAFS) analysis [Nature Nanotechnology 2019, 14, 851–857], we found out that the main bonding structures of the Joule heated Pt @CNF (salt concentration from 0.1-0.01 $\mu\text{mol}/\text{cm}^2$) are Pt-Pt and Pt-C bond. In this paper, we used a similar carbon substrate and Pt loading (0.25 $\mu\text{mol}/\text{cm}^2$). Thus, we believe the current system is also composed of mainly Pt-Pt and Pt-C bonds.

Figure R3. EXAFS spectrum of the Pt cluster and Pt single atom samples. At ultra-low Pt-salt loading (0.01 $\mu\text{mol}/\text{cm}^2$), it only shows Pt-C bonding, meaning Pt dispersed as single atoms. Higher loadings show the co-existence of Pt-Pt metal bonding and Pt-Carbon bonding. (Nature Nanotechnology 2019, 14, 851–857)

With the above evidence, we simplified our graphite substrate model by only introducing carbon dangling bond. Since there is still a trace amount of N and O, future efforts are needed to elucidate their roles on the stabilization of nanoparticles on the carbon-based substrates.

Following the Reviewer’s comment, we also compared the cases of the Pt particle on different edge locations (armchair and zigzag edge plane) to verify the strong bonding (Figure R4). There clearly exists significant electron transfer between the Pt particle and different graphene edges, which means the main interaction is strong covalent bonds rather than the van der Waals force (Figure R2b).

Figure R4 Charge density of Pt particle on different types of graphene edges: (a) armchair edge and (b) zigzag edge. The different edges are indicated by red lines. Yellow and blue regions represent the gain and loss of electrons and the scale level is set as $\pm 0.015e$.

Revision in the manuscript:

Figure R4 has been added to supporting information as Figure S10 and the following has been added to the revised manuscript on page 12:

“Considering the different orientation of graphene edges, we compared the charge densities of Pt particles on both armchair and zigzag edges (Figure S10). There exists large amount of electron transfer in both cases, which means the main interaction is strong covalent bonding rather than the van der Waals force.”

4- How can your plot of energy vs time of relaxation of the cluster show an increasing energy from 5 to 120 ps? Shouldn't it be relaxing and lowering energy? is this an increase due to atoms leaving the cluster?

Reply to the Reviewer: We thank the reviewer for pointing out this. To illustrate the stability more clearly, we calculated the potential energy curve of the Pt particle as the function of relaxation time (Figure R1). The decrease of potential energy is due to the atoms leaving the particle. But no more atoms left after 60ps and we can conclude that the particle becomes stable, as shown in Figure R1 (red curve).

5 – From you discussion of differences between the two heater systems, there should also be a discussion if thermo or electromigration could influence the process, as your observations are done under presumably high of several microamperes in a nanoscale fiber (though still It may be on too short time scale for this to be effective).

Reply to the Reviewer: We thank the reviewer for the comment. Below we discussed the possible roles of thermal heat and electromigration to the whole process of particle dispersion and stabilization.

In order to rule out the effect of electromigration on the formation of nanoparticles, we performed control experiment where electromigration was not present during synthesis. Here, we used rapid radiative heating (only high temperature with ultrafast ramping rate($>10^5$ K/s), and no direct electron current involved) was used to synthesis nanoparticles.

The rapid radiative heating setup is schematically shown in Figure R5a. A hollow cylinder connected with electrodes on both ends is made by a large piece of CNF film, where inside the cylinder, a salt coated CNF (S-CNF) film is placed. In this configuration, the outside carbon film acts as a flash high temperature radiation source to heat the S-CNF when electrical power is applied. After rapid radiation heating treatment, the prepared Pt NPs@CNF specimen was analyzed via HRSTEM. As we can see from Figure R5b, a uniform distribution of small nanoparticles were observed on the CNF surface. Similar to the Joule heating case, the amorphous carbon film showed evidence of crystallization and the NPs were associated with the T-graphite edge planes.

Figure R5. (a) Schematic image of rapid radiative heating setup, and (b) high-resolution ABF image of Pt NPs@CNF synthesized through rapid radiative heating.

To summarize, we have included the above experiments in the Table R1. Overall, it is evident that the particle formation and stabilization are governed by the rapid heating process, and not necessarily electromigration.

Table R1. A comparison of NPs synthesized through different heating methods

Heating method	Rate	Electric current pass through?	Particle size	CNF substrate
Joule heating shock	Ultrafast(>10 ⁵ K/s)	Yes	small	crystalline
Radiative heating shock	Ultrafast(>10 ⁵ K/s)	No	small	crystalline
Radiative heating slow	Fast (200K/s)	No	large	amorphous

Revision in the manuscript:

Figure R5 and Table R1 are now added to the Supporting Information as Figure S5 and Table S1. The following changes has been added in the manuscript on Page 11:

In order to rule out the effect of electromigration on the formation of nanoparticle, rapid radiative heating process was used to synthesis nanoparticles on carbon substrate. In other words, the salt-loaded CNFs were only exposed to high temperatures under ultrafast ramping rate (>10⁵K/s), and no direct electrical current was involved.

The rapid radiative heating setup is schematically shown in Figure S5a. A hollow cylinder connected with electrodes on both ends is made by a large piece of CNF film, where inside the cylinder, a salt coated CNF (S-CNF) film is placed. In this configuration, the outside carbon film acts as a flash high temperature radiation source to heat the S-CNF when electrical power is applied. After rapid radiation heating treatment, the prepared Pt NPs@CNF specimen was

analyzed via HRSTEM. As we can see from Figure S5b, a uniform distribution of small nanoparticles was observed on the CNF surface. Similar to the Joule heating case, the amorphous carbon film showed evidence of crystallization and the NPs were associated with the T-graphite edge planes. To summarize, we have included the above experiments in the Table S1. Overall, it is evident that the particle formation and stabilization are governed by the rapid heating process, and not necessarily electromigration.

6 the recent paper in nature nanotech 13 p 856-861 appears to provide an detailed description of the presumed cause of the stability. In your paper you merely state that it is stable, while still implying single Pt stability. What makes the Pt stable in those positions as in R11 – what functional groups are included in your model and what bonds are formed?

Reply to the Reviewer: We thank the reviewer for this constructive comment. In our additional calculation of potential energy (Figure R1), we can conclude the Pt particle is stable on the edge plane of graphene. The charge density calculation shows the much stronger covalent bond formation between the Pt particle and the edge plane of graphene. The covalent bond leads to the lower potential energy of the Pt particle and strongly bind the Pt particle and the edge plane. The content of other elements in our experiments is very low based on our quantitative STEM-EDS results (Figure 3d). It can be seen that N and O drop tremendously after Joule heating, from the initial 13.0 wt% and 4.3 wt% to the final 1.7 wt% and 1.2 wt%, respectively. Thus, we did not consider the functional groups in our MD simulation. While we have to passivate part of boundary carbon by hydrogen to equilibrate the electron in the DFT calculation, due to the electron coupling method, it has been confirmed that it will not influence the interaction between the graphene and the particle and is widely used in other researches of graphene flake [*The Journal of Physical Chemistry C*, 117 (2013) 25424-25432]. Through our DFT calculation, we confirm the bonding between the Pt particle and the graphene edge planes are covalent bond type, while the main interaction between the Pt particle and the graphene basal planes is van der Waals forces.

Figure 3d. Quantitative EDS analysis of the CNF before and after Joule heating.

Revision in the manuscript:

The following has been added to the revised manuscript on page 18.

“The content of other elements in our experiments is very low (Figure 3d). Thus, the functional groups were not considered in our MD simulation.”

7 *what terminating functional groups are used in the simulation and what would be relevant to use? These could considerably influence the systems, e.g. –H vs –OH on the edge planes.*

Reply to the Reviewer: The content of other elements in our experiments is very limited (Figure 3d). Thus, we did not consider the functional groups in our MD simulation. While we have to passivate part of boundary carbon by hydrogen to equilibrate the electron in the DFT calculation, due to the electron coupling method, it has been confirmed that it will not influence the interaction between the graphene and the particle and is widely used in other researches of graphene flake [*The Journal of Physical Chemistry C*, 117 (2013) 25424-25432].

8 - *In your long time stability test, it is not clear how the ex situ formed sample was formed – which should be clarified, and also lead on to the question how you foresee this synthesis method can be scaled up to any reasonable amount of relevance? Unless of course you provide more detailed chemical insight to the underlying cause of the stability in the above questions that chemists then can try to replicate.*

Reply to the reviewer: We thank the reviewer for the comment. Below, we provide further information on the ex-situ synthesis method and the approaches for scale up.

1. Ex-situ Joule heating sample preparation. The ex-situ sample was prepared through Joule heating of salt loaded CNF film (schematically shows in Figure R6), where the CNF film image can be seen from Figure R7 and the loading amount of Pt salt precursor was $0.25 \mu\text{mol}/\text{cm}^2$.

Figure R6. Schematic of ex-situ Joule heating setup.

2. Scale up methods. We also show several examples that could easily lead to the scalable synthesis of nanoparticles using this emerging technique.

- ❖ The first direct scale up method is to make use of a large piece of CNF film. Therefore, the dimension of this method goes from *nm* level (single CNF) to *mm* level (CNF film, Figure R7).

Figure R7. Scale up with a piece of CNF film. (a) millimeter-scale CNF film. (b) SEM image of the CNF film

- ❖ With a proper design of the Joule heating setup, the scalability can increase several times by Joule heating many CNF films at one time. As demonstrated in figure R8, CNF films were connected in series in a home-made setup (Figure R8a) and can be ramped to high temperature at the same time during Joule heating (Figure R8b).

Figure R8. Scale up the rapid Joule heating synthesis method by aligning several CNF films in one circuit. (a) a home-made setup connecting six CNF films in one circuit. (b) Heating all the CNF films simultaneously through Joule heating [*Proceedings of the National Academy of Sciences* 117, 6316-6322 (2020)].

- ❖ Large scale and 3D bulk samples using direct Joule heating
We have demonstrated previously the Joule heating on a large piece of carbonized wood (Figure R9a) [*Science* 2018 359(6383):1489-1494]. By making use of this 3D bulk carbon substrate, uniform nanoparticles can be synthesized on the surface (Figure R9b) after rapid Joule heating.

Figure R9. Scale up on a 3D carbonized wood with uniform nanoparticles [*Science* 2018 359(6383):1489-1494]

- ❖ Lastly, the most recent literature publication [*Nature* 2020, 577, 647-651] also used rapid Joule heating method and showed its scalability: Gram level graphene/graphite material can be synthesized through ultrahigh temperature Joule heating in milliseconds from amorphous carbon sources. In the meantime, our work also provides direct visualization and detailed elucidation of the carbon material graphitization process during high temperature electrical Joule heating process, which is expected to raise tremendous attention from the readers.

Revision in the manuscript:

Figure R6 and R7 have been added to the supporting information as Figure S16 and Figure S17, respectively. The following changes has been made on Page 17:

The ex-situ sample was prepared through Joule heating of salt loaded CNF film (schematically shows in Figure S16), where the CNF film image can be seen from Figure S17 and the loading amount of Pt salt precursor was $0.25 \mu\text{mol}/\text{cm}^2$.

The following changes has been made on Page 16:

This method can be scaled up. The process is not limited to single CNF, and in fact large sheets of CNF films can be subjected to Joule heating (specimen dimensions from *nm* (single CNF) to *mm* level (CNF film) as shown in Figure S17. This process can be further scaled up by connecting several CNF films in series where electrical current pass through the films⁴⁹. In addition, by switching the carbon substrate to a bulk carbonized wood²⁰, the scalability increases to tens of centimeters. A recent study⁵⁰ also demonstrated that rapid Joule heating method is highly scalable, where gram level graphene/graphite material was synthesized through ultrahigh temperature Joule heating in milliseconds from amorphous carbon sources. In the meantime, our work also provides direct visualization and detailed elucidation of the carbon material graphitization process during high temperature electrical Joule heating process.

Reviewer #2 (Remarks to the Author):

On page 261 -> "density functional theory (DFT).

The authors improved the simulations and the discussion of the results that they presented in the previous version of the paper. The discussion about the pinning mechanisms could be more polished though.

What I guess, from experimental and simulational results: larger particles (how big should they be?) are anchored to graphene edges through Pt atoms that intercalated into the planes. We should therefore have three distinct regions to consider: one in which we basically have graphene with a few, isolated Pt atoms intercalated, far from the edges; a second one quite rich in Pt, forming a stable second phase (a graphene-Pt alloy?), near the edges; and the Pt particle itself, attached to the Pt-rich edges through Pt-Pt and Pt-C bonds strong enough to impede particle glide. On the other hand, the small particles, the ones that eventually disappeared, were too small and got dissolved by continuously losing atoms that diffused into the graphene planes.

In any case, full elucidation of these mechanisms would require larger and longer simulations and systems of different sizes.

Reply to the Reviewer: We thank the reviewer for this constructive comment. We calculated the charge density of Pt on three distinct regions by the DFT method. Figure R10a shows the isolated Pt atoms intercalated far from the edge plane. The Pt atoms prefer to locate on the bridge site of graphene. Similar to the Pt particle on the basal plane, there exists few charge transfer between Pt and C atoms (Figure R10d). While the Pt atoms at the edge plane have the regular charge transfer and are bonded with graphene through covalent bonds. Thus, it forms the graphene-Pt alloy (Figure R10b and e). When the Pt particle is attached to the Pt-rich edges (Figure R10c and f), the charge transfer becomes irregular and the change of charge density leads to the stability of particle at the edge plane.

Figure R10 Atomistic structure of (a) isolated Pt atoms intercalated graphene, (b) Pt-graphene alloy and (c) Pt particle on the edge plane of Pt-graphene alloy. Brown, silver and pink balls

represent the C, Pt and H atoms, respectively. (d)~(f) Corresponding charge density. Yellow and blue regions represent the gain and loss of electrons and the scale level is set as $\pm 0.015e$.

Furthermore, we also calculated the Pt particle with smaller sizes, which is constructed with 12, 25, 50 and 75 atoms, adsorbed on the edge plane of graphene (Figure R11). It can be found that the Pt₁₂, Pt₂₅ and Pt₅₀ disappeared and the Pt atoms intercalated into the graphene layers, while Pt₇₅ is stable after parts of atoms intercalated into graphene layers. It can be concluded that there are not enough Pt-Pt bonds in the small Pt particle to resist the Pt intercalation, leading to the disappearance of Pt particle. When the particle is large enough (number of atoms (n) in our simulation is larger than 75), remaining Pt-Pt bonds are strong enough to resist further intercalation. Figure R5e shows the potential energy of Pt particles as the function of relaxation time. All these curves converge to a constant. The smaller the particle is, the more quickly the curve converges. It also means the intercalated Pt atoms are stable in the graphite layers, even the particle disappeared (n<75). On the other hand, it confirmed that some small particles disappeared, and only large particles can be maintained in our experiments.

Figure R11 The morphological change of Pt particle in the presence of T-graphene with edge plane: (a) Pt₁₂, (b) Pt₂₅, (c) Pt₅₀ and (d) Pt₇₅. (e) The potential energy of Pt particles with different sizes as the function of relaxation time.

Revision in the manuscript:

Figure R10 and R11 have been added to the supporting information as Figure S11 and S15, respectively. And the following has been added to the revised manuscript on page 12:

“Furthermore, three distinct regions are considered to analyze the stability of Pt. Figure S11a shows the isolated the Pt atoms intercalated far from the edge plane. The Pt atoms prefer to locate on the bridge site of graphene. Similar to the Pt particle on the basal plane, there exists

few charge transfer between Pt and C atoms (Figure S11d), while the Pt atoms at the edge plane have the regular charge transfer and are bonded with graphene through covalent bonds. Thus, it forms the graphene-Pt alloy (Figure S11b and e). When the Pt particle is attached to the Pt-rich edges (Figure S11c and f), the charge transfer becomes irregular and the change of charge density leads to the stability of particle at the edge plane.”

The following has been added to the revised manuscript on page 15:

“To verify the assumption, we also calculated the Pt particle with smaller sizes, which is constructed with 12, 25, 50 and 75 atoms, adsorbed on the edge plane of graphene (Figure S15). It can be found that the Pt₁₂, Pt₂₅ and Pt₅₀ disappeared and the Pt atoms intercalated into the graphene layers, while Pt₇₅ is stable after parts of atoms intercalated into graphene layers. It can be concluded that there are not enough Pt-Pt bonds in the small Pt particle to resist the Pt intercalation, leading to the disappearance of Pt particle. When the particle is large enough (number of atoms (n) in our simulation is larger than 75), remaining Pt-Pt bonds are strong enough to resist further intercalation. Figure S15e shows the potential energy of Pt particles as the function of relaxation time. All these curves converge to a constant. The smaller the particle is, the more quickly the curve converges. It also means the intercalated Pt atoms are stable in the graphite layers, even the particle disappeared (n<75). On the other hand, it confirmed that some small particles disappeared, and only large particles can be maintained in our experiments.”

REVIEWERS' COMMENTS

Reviewer #3 (Remarks to the Author):

As a substitute to the referee 1, who extensively commented and thoroughly reviewed the manuscript, I find that rebuttals now addresses the comments raised by the referee during both rounds. I have no further questions or comments.